# FlexiVoice: Enabling Flexible Style Control in Zero-Shot TTS with Natural Language Instructions

**Dekun Chen[1], Xueyao Zhang[1], Yuancheng Wang[1], Kenan Dai[2], Li Ma[2], Zhizheng Wu[1345]**

[1]The Chinese University of Hong Kong, Shenzhen [2]Huawei Technologies Co., Ltd
[3]Shenzhen Loop Area Institute [4]City University of Macau [5]Amphion Technology Co., Ltd.

## Abstract

This study proposes *FlexiVoice*, a text-to-speech (TTS) synthesis system capable of flexible style control with zero-shot voice cloning. The speaking style is controlled by a natural-language instruction and the voice timbre is provided by a speech reference in zero-shot manner. FlexiVoice is built with an LLM core, which takes text as input, and also takes an optional natural language instruction and an optional speech reference to control style and timbre, respectively. FlexiVoice is equipped with a novel **Progressive Post-Training (PPT)** scheme that progressively unlocks accurate and flexible controllability. In particular, it first employs Direct Preference Optimization (DPO) to enable FlexiVoice to accurately follow both natural language instruction and speech reference simultaneously. It then uses a multi-objective Group Relative Policy Optimization (GRPO) to disentangle style instruction, reference timbre, and textual content. Finally, it adapts instruction GRPO for more advanced instruction following. Experimental results show that FlexiVoice surpasses competing baselines and demonstrates strong capability in decoupling control factors. Human evaluations further confirm its naturalness, controllability, and robustness. Audio samples are available at https://flexi-voice.github.io/.

## 1 Introduction

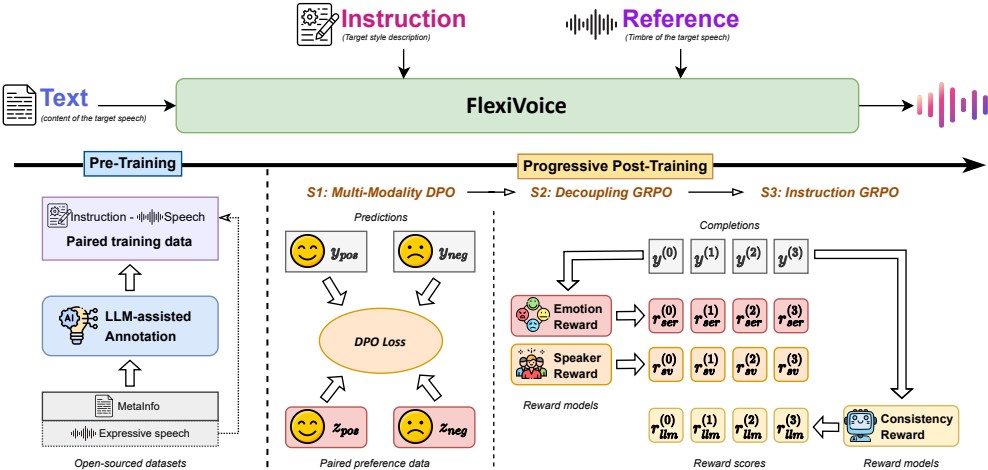

Figure 1: An overview of FlexiVoice that supports diverse style generation with arbitrary voice timbres. It takes an optional natural language **instruction** for style and an optional **reference** speech for timbre. It consists of Pre-Training and Progressive Post-Training (PPT) stages. The PPT process includes three processes, **S1:** Multi-modality DPO, **S2:** Decoupling GRPO, **S3:** Instruction GRPO.

Recent advances in text-to-speech (TTS) have been largely driven by the emergence of Large Language Models (LLMs) and refined post-training techniques. A notable breakthrough is zero-shot TTS (Du et al., 2024; Zhou et al., 2025), which enables voice cloning with only a short reference speech, effectively capturing and reproducing a speaker's timbre. Beyond timbre, controlling speaking style has become an important challenge. One direction, exemplified by Vevo (Zhang et al., 2025c) and IndexTTS2 (Zhou et al., 2025), employs two separate speech references to control timbre and style separately. Another line of work, instruction-based TTS (Vyas et al., 2023; Zhou et al., 2024b; Ji et al., 2024b), leverages natural language instructions to specify the target style. However, existing instruction-driven models often struggle either to faithfully follow the instructions or to maintain stable timbre consistency.

Achieving flexible style control in zero-shot TTS presents a unique challenge: the '**Style-Timbre-Content Conflict**'. In standard supervised training, models tend to over-rely on the strong acoustic priors from the reference speech (timbre leakage) or infer prosody from the text (content leakage), often ignoring the explicit style instruction. Merely applying instruction conditioning is insufficient to resolve these entangled modalities. Therefore, a robust framework is required not just to condition the model, but to actively decouple these factors and enforce instruction adherence against conflicting acoustic cues.

In this work, we propose **FlexiVoice**, a TTS system that can flexibly control the speaking style with a zero-shot voice. In particular, it can take a natural language instruction and a reference speech or one of them for flexible controllability. The natural language instruction aims to control speaking styles (e.g. emotion, speaking speed) and the speech reference is to control timbre for speaker identity. FlexiVoice is built on top of a pre-trained large language model (LLM). The LLM core equips FlexiVoice with a robust and comprehensive instruction-following ability. To achieve flexible controllability, we first construct a large-scale and diverse speech dataset with natural language instructions, named **FlexiVoice-Instruct**. The FlexiVoice-Instruct dataset is annotated with the help of LLM. We then pre-train FlexiVoice with Emilia (He et al., 2024) and FlexiVoice-Instruct. In post-training, we propose a novel **Progressive Post-Training (PPT)** framework. Unlike general LLM alignment, PPT is specifically designed to resolve the modality conflicts in TTS through a systematic curriculum. It consists of three stages: (1) **Multi-modality DPO** establishes the initial alignment for instruction adherence; (2) **Decoupling GRPO** introduces a multi-objective optimization with conflicting data scenarios to mathematically enforce the separation of style from timbre and content; and (3) **Instruction GRPO** leverages an audio-language model (ALM) reward to generalize this capability to complex, open-ended instructions. This progressive formulation transforms the instruction-following task from simple conditioning into a rigorous disentanglement process.

We evaluate FlexiVoice from flexible controllability and instruction-following ability aspects, using emotion datasets and the InstructTTSEval (Huang et al., 2025) benchmark. The experimental results show that FlexiVoice can decouple speaking style (e.g. emotion) and speaker identity. In comparison with baselines, it achieves large gains in instruction adherence and robustness on the multi-modality control evaluation, and demonstrates strong performance on complex instruction tasks. Human evaluation results further confirm the naturalness and robustness of the generated speech.

The contributions are summarized as follows:

- We propose **FlexiVoice**, a TTS system that enables flexible and precise style control for zero-shot voice synthesis. The speaking style is controlled by an optional natural language instruction and the timbre is guided by an optional reference speech. It enables any combination of style and timbre for flexible control.

- We develop a large-scale and diverse speech dataset with natural language instructions, **FlexiVoice-Instruct**. The annotations are created using an LLM (i.e. Deepseek-V3 (Liu et al., 2024a)) based on the speech properties. It has a broad coverage of human-like instructions, including expressive scenarios.

- We propose a **Progressive Post-Training (PPT)** framework, a novel curriculum learning approach tailored for controllable TTS. By strategically sequencing preference alignment and multi-objective optimization, PPT explicitly solves the Style-Timbre-Content conflict, achieving robust disentanglement that standard training paradigms fail to handle.

## 2 RELATED WORK

**Controllable Speech Synthesis**   Research on controllable speech synthesis has mainly followed two directions. (i) *Zero-shot TTS* clones timbre from short reference speech (Chen et al., 2024; Wang et al., 2024; Zhou et al., 2025; Zhang et al., 2025c), while partially controlling speaking style. (ii) *Instruction-based TTS* uses natural-language prompts to specify style; for instance, PromptTTS (Guo et al., 2023) and PromptStyle (Liu et al., 2023) enable text-guided style control but within limited spaces, and Parler-TTS (Lyth & King, 2024) scales conditioning with weak labels without supporting timbre control. More recent systems, such as VoxInstruct (Zhou et al., 2024b), AudioBox (Vyas et al., 2023), ControlSpeech (Ji et al., 2024b), and CosyVoice2 (Du et al., 2024), combine instruction with reference speech, but still lack robust disentanglement and broad style diversity. These limitations highlight the need for a unified framework that processes multi-modal inputs, generates speech following instruction-defined style, preserves timbre, and explicitly addresses disentanglement.

**Instruction-Speech Dataset**   Instruction-based TTS has driven datasets that couple text descriptions with speech. TextrolSpeech (Ji et al., 2024a) introduced large prompt–speech pairs with five style factors; Audiobox (Vyas et al., 2023) broadened the paradigm to multi-modal audio generation, though its lack of public release limited accessibility. Parler-TTS (Lyth & King, 2024) scaled weak labels for speaker/style/conditions to tens of thousands of hours. SpeechCraft (Jin et al., 2024) and ParaSpeechCaps (Diwan et al., 2025) enriched granularity with automatic captioning and diverse paralinguistic attributes. Nonetheless, most corpora come from homogeneous sources and emphasize templated descriptions, leaving insufficient coverage of natural, diverse instructions. Our dataset targets higher-quality, more natural annotations to strengthen instruction-following.

**Reinforcement Learning in Speech Synthesis**   Reinforcement learning has recently been explored to improve controllability in speech synthesis. INTP (Zhang et al., 2025a) applies DPO to challenging zero-shot cases for better intelligibility. Emo-DPO (Gao et al., 2025) extends preference alignment to emotional control. Vevo2 (Zhang et al., 2025b) employs multi-objective post-training to jointly enhance intelligibility and prosody across both speech and singing. These works highlight the promise of RL-based alignment for targeted aspects of TTS. In contrast, our approach adopts a progressive curriculum that leverages reinforcement learning to explicitly address modality disentanglement and complex instruction following, thereby enabling broader controllability in zero-shot multi-modality TTS.

## 3 OVERVIEW OF FLEXIVOICE

**FlexiVoice** is built on top of an LLM similar to other recent TTS systems (Du et al., 2024; Zhou et al., 2025). In FlexiVoice, a speech tokenizer converts speech into discrete tokens as inputs and outputs of an LLM. The LLM core processes the input text, natural-language instruction, and reference tokens to generate discrete speech tokens. The generated tokens are transformed into Mel-spectrogram features via flow matching (Lipman et al., 2023), and finally converted into waveform audio with a vocoder. A detailed description is provided in Appendix A.1. FlexiVoice is first pre-trained on a large-scale dataset as **FlexiVoice-Base** and then post-trained using a progressive strategy.

### 3.1 PRE-TRAINING

**FlexiVoice-Base** is a pre-trained model of FlexiVoice. It is pre-trained using Emilia (He et al., 2024) and a diverse set of instruction speech dataset as listed in Appendix A.2.

Emilia (He et al., 2024) is a large-scale, multilingual, and diverse speech dataset that covers a broad spectrum of speaking styles and scenarios, and primarily supports the fundamental ability of speech generation. To further enable instruction-guided TTS, we construct a high-quality and diverse dataset, **FlexiVoice-Instruct**, consisting of natural-language instructions across various scenarios. The processing pipeline is detailed in Section 4. In addition, we enrich the pre-training phase with existing instruction–speech corpora (Diwan et al., 2025) and specialized resources featuring attributes such as emotion, age, and debate, with processing details provided in Appendix A.2. We also incorporate NVSpeech (Liao et al., 2025), which provides paralinguistic tags and expressive coverage. Together, these resources form the pre-training corpus for our model.

Following the traditional pre-training strategy for TTS systems (Du et al., 2024; Zhou et al., 2025), we only train the LLM core and keep other modules frozen, without incorporating the reference speech during pre-training. As shown in Figure 3, the text and instruction are first formatted by the LLM input template. The paired ground-truth speech is preprocessed by a frozen speech tokenizer into discrete tokens, which are used to compute loss with generated tokens during pretraining. To ensure consistency in input format, we apply *Speak the following text* as the default instruction when encountering data from Emilia (He et al., 2024) and NVSpeech (Liao et al., 2025) that lacks an explicit instruction.

## 3.2 POST-TRAINING

We propose a **Progressive Post-Training (PPT)** scheme to disentangle speaking style and timbre and to enable FlexiVoice to have complex instruction-following ability. It is inspired by Curriculum Learning (Bengio et al., 2009), which starts from simpler objectives and gradually advances to harder ones for stable optimization and better generalization. After pre-trianing, FlexiVoice-Base has solid zero-shot TTS capability but still struggles with multi-modality inputs and complex instructions. PPT is to advance FlexiVoice-base for generalization and better performance. This progressive curriculum ultimately yields a robust multi-modality instruction TTS model, **FlexiVoice**.

PPT has three stages:

- **S1** aligns multi-modality controllability of instruction and reference speech in controlled emotion-centric tasks with explicit labels.
- **S2** disentangles the timbre and style in reference speech, and the content and style in target text, within the same scenario as S1,
- **S3** extends to complex real-world instructions that are more ambiguous and harder to align.

### 3.2.1 S1: MULTI-MODALITY CONTROLLABILITY

S1 mainly employs DPO to empower FlexiVoice processing style instruction and timbre reference at the same time. The first two stages target emotional instructions to reduce task complexity and explicitly address the core issue. We restrict instructions to templates like *Use {label} emotion to read it*, with labels chosen from *Neutral, Happy, Angry, Sad, and Surprised*.

For emotion-related tasks, paired preference data can be directly obtained from Speech Emotion Recognition (SER) datasets. We use the Emotional Speech Dataset (ESD) (Zhou et al., 2021), where the same speaker reads identical sentences with different emotions. Following Gao et al. (2025), for each data point, we assign a target emotion label (e.g., Happy) via an instruction template (all templates are listed in Appendix A.5), select a sentence with the target emotion as the preferred sample, the identical sentence with a different emotion (e.g., Angry) as the dis-preferred one, and use a neutral sample from the same speaker as the reference speech (an example pair is provided in Appendix A.5).

DPO directly aligns the model's emotional output with the instruction and reference speech without requiring an explicit reward model (Rafailov et al., 2023). The preference dataset $\mathcal{D}$ consists of tuples $(x, y_w, y_l)$, where $x$ includes instruction, text, and reference; $y_w$ is the "winner" response that matches the instruction, and $y_l$ is the "loser." The DPO loss is defined as:

$$\mathcal{L}_{\text{DPO}}(\pi_\theta; \pi_{\text{ref}}) = -\mathbb{E}_{(x, y_w, y_l) \sim \mathcal{D}} \left[ \log \sigma \left( \beta \log \frac{\pi_\theta(y_w|x)}{\pi_{\text{ref}}(y_w|x)} - \beta \log \frac{\pi_\theta(y_l|x)}{\pi_{\text{ref}}(y_l|x)} \right) \right]$$

where $\pi_\theta$ is the policy model and $\pi_{\text{ref}}$ is the reference model, both initialized from **FlexiVoice-Base**.

### 3.2.2 S2: DECOUPLING OF REFERENCE SPEECH AND TARGET TEXT

S2 focuses on FlexiVoice's decoupling capability in scenarios where speech reference and target text contain styles conflicting with instructions. After DPO training, the model follows emotional instructions well under neutral references, but interference remains when references or texts themselves are emotion-laden, conflicted with the target emotion defined in the instruction. To explicitly suppress these effects, we adopt a multi-objective GRPO formulation by constructing conflicting training scenarios (e.g., Happy instruction vs. Sad reference). The reward $r_{ser}$ (defined below) acts

as a style constraint, penalizing the model if it leaks style from the reference or text. Conversely, $r_{sv}$ acts as a timbre constraint, ensuring speaker identity is preserved. By optimizing the joint advantage, the model is forced to decouple these factors to maximize the total reward. We use the same instruction templates, combine neutral and emotional clips from SER datasets as reference speech, and sample texts from NCSSD (Liu et al., 2024b) (details in Appendix A.4).

Rewards are defined as follows: (1) $r_{ser} \in (0, 1)$, the probability score for the instructed emotion given by the emotion recognition result from Emotion2vec-Large (Ma et al., 2024); (2) $r_{sv} \in \{0, 1\}$, the speaker verification result from CAM++ (Wang et al., 2023) to ensure timbre consistency. Following Zhang et al. (2025b), the advantage with multi-objective rewards is:

$$A_{emo}^i = \frac{r_{ser}^i - \text{mean}(r_{ser}^i)}{\text{std}(r_{ser}^i)} + \frac{r_{sv}^i - \text{mean}(r_{sv}^i)}{\text{std}(r_{sv}^i)}$$

where $i$ indexes the $i$-th completion among $K$ candidates for the same input $x$.

### 3.2.3 S3: Enhancement on Complex Instruction-following

The final stage enhances instruction following on complex, real-world directives beyond emotion tasks. Since paired preference data are infeasible at this scale, we directly employ GRPO. Unlike the first stage, where rewards are available from SER models, assessing the consistency between speech and open-ended instructions is more difficult. Huang et al. (2025) shows that Gemini-2.5-pro provides reliable judgments aligned with human preferences, but its use in GRPO is cost-prohibitive. We instead adopt the open-sourced Kimi-Audio-7B-Instruct (Ding et al., 2025) as the reward model due to its strong speech comprehension. It is prompted to output a binary yes/no decision on whether the generated speech matches the instruction, which is mapped into a reward $r_{llm} \in \{1, 0\}$.

For this stage, we use only instruction and text as inputs, discarding references, since references may conflict with open-ended constraints (e.g., gender) and destabilize training. Details of the data construction process are given in Appendix A.4.

To mitigate catastrophic forgetting, we mix a small portion of S2-GRPO data during this stage. Thus, the final GRPO training becomes a multi-task, multi-objective optimization. The single-objective advantage in this stage (left) and the final advantages (right) are:

$$A_{ins}^i = \frac{r_{llm}^i - \text{mean}(r_{llm}^i)}{\text{std}(r_{llm}^i)}, \qquad A^i = \begin{cases} A_{emo}^i & \text{for inputs in S2,} \\ A_{ins}^i & \text{for inputs in S3.} \end{cases}$$

## 4 FlexiVoice-Instruct Dataset

### 4.1 Overview

To equip the model with fundamental instruction-following capability in the pre-training phase, it is essential to build a large-scale instruction–speech dataset covering diverse styles and scenarios. We therefore construct a high-quality and diverse dataset totaling 4,316 hours, where speech-related textual metadata is processed with an LLM-based annotator. This approach enables efficient generation of natural, high-level instructions that better reflect real human usage patterns.

### 4.2 Main Process

We employ two data sources, Emilia and game voice acting, using a unified processing strategy:

Emilia (He et al., 2024) is a large-scale, diverse, in-the-wild speech dataset. Because the audio originates from video platforms and podcasts, some entries include source metadata such as video titles and tag lists. Combined with transcriptions, these cues allow us to infer speaking style and scene context, enabling the generation of open-ended instructions even without direct speech analysis. We employ Deepseek-V3 (Liu et al., 2024a) for instruction annotation. To filter noisy samples (e.g., URLs or conflicting metadata–transcription pairs), the LLM is prompted to evaluate the informational value of metadata for style and scenario inference, ensuring higher data quality.

To enhance expressiveness and stylistic richness, game voice acting data from two popular games is additionally incorporated. A distinctive feature of this data is the strong link between speaking

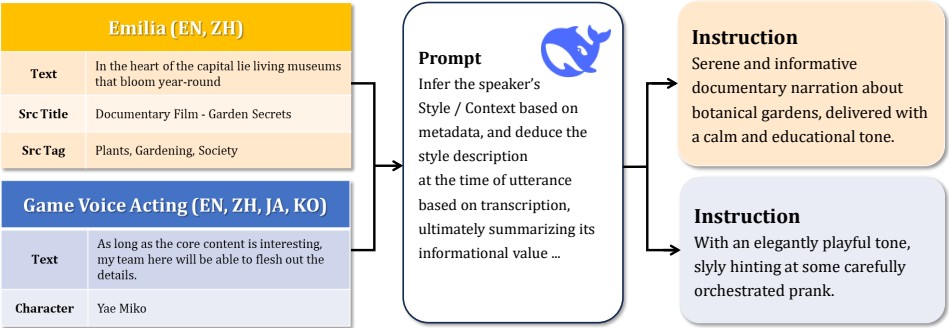

Figure 2: Processing flow and examples for FlexiVoice-Instruct. For both sources, we use the speech transcription and related meta information to prompt LLM to generate natural and human-like descriptions, as the instruction in our pre-training stage.

style and character personality. LLMs, trained on large-scale web corpora, can often recognize these characters and capture their iconic styles. We thus use Deepseek-V3 (Liu et al., 2024a) to generate instructions conditioned on both transcription and character name. Concretely, we first supply the full character list and let the LLM identify those it can reliably recognize. For known characters, we prompt the LLM to infer personality and speaking style from the name, then refine the description using the transcription. As with Emilia, we incorporate an informational value check to filter noisy annotations. Prompts are provided in Appendix A.3, while the overall processing flow and examples are shown in Figure 2.

## 5 EXPERIMENTS

### 5.1 EXPERIMENTAL SETUP

**Benchmarks** To evaluate the model's multi-modality controllability as well as its capacity to follow complex real-world instructions, we employ one self-constructed dataset and one open-source benchmark. For multi-modal control and disentanglement, we define two task types with two difficulty levels each, focusing on instruction-based emotional TTS, as summarized in Table 1. For complex instruction following, we adopt the InstructTTSEval (Huang et al., 2025) benchmark.

Table 1: Tasks for multi-modality control and disentanglement evaluation. There are two tasks with two difficulties, with an example for each task.

| Task Type | Difficulty | Text | Reference | Example |
|---|---|---|---|---|
| Single Modality: Text-Only (TO) | Easy | Neutral | - | Instruction: Speak it using **happy** emotion. Text: Today is Monday. Reference Speech: - |
| | Hard | Emotional | - | Instruction: Speak it using **happy** emotion. Text: I'm so **sad** that it's raining outside. Reference Speech: - |
| Multi-Modality: Text and Reference (TR) | Easy | Neutral | Neutral | Instruction: Speak it using **happy** emotion. Text: Today is Monday. Reference Speech: [A **neutral** voice] |
| | Hard | Neutral | Emotional | Instruction: Speak it using **happy** emotion. Text: Today is Monday. Reference Speech: [A **sad** voice] |

The first evaluation set is built from MEAD (Wang et al., 2020) (English) and CSEMOTIONS (Tian et al., 2025) (Chinese). In TR tasks, reference speech is randomly selected as either neutral or conflicting emotional clips from the same speaker as the ground-truth. For TO-hard, the text is replaced with sentences carrying emotions different from the target, so no ground-truth audio exists for this case. We retain five emotion categories from both datasets and randomly sample 500 examples each for English and Chinese, ensuring balanced coverage of target emotions.

Table 2: Multi-modality controllability and disentanglement evaluation result in different tasks and two languages.

| Model | Text-Only as input (TO) | | | | Text and Reference Speech as input (TR) | | | | | | |
|---|---|---|---|---|---|---|---|---|---|---|---|
| | Easy | | Hard | | Easy | | | Hard | | | |
| | ACC-I↑ | E-SIM↑ | ACC-I↑ | ACC-T↓ | ACC-I↑ | E-SIM↑ | SV↑ | ACC-I↑ | ACC-R↓ | E-SIM↑ | SV↑ |
| EN | | | | | | | | | | | |
| Groud-truth | 93.4 | 1.00 | - | - | 93.4 | 1.00 | - | 93.4 | 0.6 | 1.00 | - |
| Parler-TTS | 44.6 | 0.72 | 12.2 | 42.0 | - | - | - | - | - | - | - |
| PromptStyple | 43.8 | 0.70 | 14.0 | 33.6 | - | - | - | - | - | - | - |
| PromptTTS | 57.8 | 0.79 | 15.0 | 41.0 | - | - | - | - | - | - | - |
| CosyVoice2 | - | - | - | - | 65.6 | 0.85 | **99.8** | 61.0 | 14.4 | 0.84 | **99.8** |
| VoxInstruct | 70.6 | 0.84 | 17.8 | 41.2 | 58.5 | 0.81 | 89.0 | 49.7 | 23.9 | 0.80 | 90.6 |
| **FlexiVoice-Base** | 72.4 | 0.83 | 39.4 | 30.6 | 58.8 | 0.81 | 99.2 | 48.8 | 32.2 | 0.78 | 99.4 |
| **FlexiVoice** | **97.4** | **0.89** | **89.4** | **6.6** | **89.4** | **0.90** | 91.0 | **78.2** | **10.6** | **0.87** | 95.8 |
| ZH | | | | | | | | | | | |
| Groud-truth | 61.6 | 1.00 | - | - | 61.6 | 1.00 | - | 61.6 | 4.4 | 1.00 | - |
| CosyVoice2 | - | - | - | - | 44.4 | 0.84 | **99.8** | 47.8 | 15.3 | 0.79 | **100.0** |
| VoxInstruct | 48.6 | **0.76** | 29.0 | 21.2 | 19.4 | 0.75 | 46.8 | 18.7 | 23.2 | 0.73 | 59.8 |
| **FlexiVoice-Base** | 78.4 | **0.76** | 66.8 | 14.2 | 25.2 | 0.78 | 99.6 | 22.4 | 38.0 | 0.74 | 99.2 |
| **FlexiVoice** | **99.8** | 0.72 | **98.4** | **0.8** | **81.8** | **0.85** | 98.8 | **75.8** | **13.2** | **0.80** | 98.4 |

The second benchmark, InstructTTSEval (Huang et al., 2025), serves as a comprehensive test set for complex instruction following beyond simple emotions. It comprises 1,000 English and 1,000 Chinese samples covering 12 distinct speech attributes: *gender, pitch, texture, clarity, fluency, speed, accent, age, volume, emotion, tone, and personality*. These attributes are encapsulated in instructions across three task levels—Acoustic-Parameter Specification (APS), Descriptive-Style Directive (DSD), and Role-Play (RP)—allowing us to rigorously evaluate the model's versatility in handling diverse and fine-grained style controls.

**Metrics** For the first evaluation set, we use Emotion2vec-Large (Ma et al., 2024) for SER and emotion embedding extraction. We report instruction-following accuracy with respect to the target emotion (**ACC-I**, ↑), and in the hard settings also report accuracy against conflicting emotions from text or reference (**ACC-T, ACC-R**, ↓), where lower is better. Cosine similarity between emotion embeddings of ground-truth and generated speech (**E-SIM**, ↑) further reflects adherence. To verify timbre preservation in TR tasks, we use CAM++ (Wang et al., 2023) for speaker verification accuracy (**SV**, ↑). We also evaluate intelligibility and perceptual quality: word error rate for English (**WER**, ↓, ASR by Whisper-Large-V3) and character error rate for Chinese (**CER**, ↓, ASR by Paraformer-zh), speech quality MOS (**Q-MOS**, 1–5, ↑), and Comparative MOS (**CMOS**, –2 to 2, ↑). Details of the subjective setup are provided in Appendix A.9.

For complex instruction following, we follow InstructTTSEval (Huang et al., 2025), which uses Gemini as the judge model to assess consistency, and report macro-average accuracy.

**Baselines** We compare FlexiVoice against representative open-source instruction TTS systems, including Parler-TTS (Lyth & King, 2024), reproducible variants (Ji et al., 2024b) of Prompt-Style (Liu et al., 2023) and PromptTTS (Guo et al., 2023), VoxInstruct (Zhou et al., 2024b), and CosyVoice2 (Du et al., 2024), which jointly processes natural-language instructions and reference speech. In addition, we report results from InstructTTSEval (Huang et al., 2025), which cover both the above open-source systems and closed-source commercial models (gemini-2.5-flash-preview-tts, gemini-2.5-pro-preview-tts, gpt-4o-mini-tts, and Hume), and we further include MiMo-Audio-7B-Instruct, a recent large audio-language model with strong instruction-following capability.

## 5.2 DECOUPLING ABILITY OF FLEXIVOICE

From Table 2, we observe that FlexiVoice demonstrates strong capabilities in handling multi-modality inputs and disentangling content, timbre, and style across both English and Chinese tasks. The discussion is organized into four perspectives.

Table 3: WER scores and subjective evaluation results in the first evaluation set.

| Model | Text-Only as input (TO) | | | Text and Reference Speech as input (TR) | | | | | |
|---|---|---|---|---|---|---|---|---|---|
| | Easy | | | Easy | | | Hard | | |
| | WER↓ | Q-MOS↑ | CMOS↑ | WER↓ | Q-MOS↑ | CMOS↑ | WER↓ | Q-MOS↑ | CMOS↑ |
| EN | | | | | | | | | |
| Groud-truth | 4.50 | 3.16 ±0.07 | 0.00 | 4.50 | 3.50 ±0.13 | 0.00 | 4.50 | 4.26 ±0.22 | 0.00 |
| CosyVoice2 | - | - | - | 3.71 | 3.50 ±0.36 | -0.75 ±0.32 | 3.60 | 3.68 ±0.43 | -0.88 ±0.21 |
| VoxInstruct | 7.29 | 3.02 ±0.34 | -0.75 ±0.44 | 14.44 | 2.10 ±0.23 | -1.50 ±0.19 | 12.61 | 2.66 ±0.63 | -0.86 ±0.36 |
| **FlexiVoice-Base** | **5.01** | 3.72 ±0.14 | -0.12 ±0.39 | 5.31 | **3.90** ±0.26 | -1.25 ±0.29 | 6.55 | 3.82 ±0.29 | -0.56 ±0.31 |
| **FlexiVoice** | 5.99 | **4.08** ±0.29 | **0.91** ±0.20 | 5.23 | 3.62 ±0.25 | **0.89** ±0.30 | 6.99 | **4.06** ±0.35 | **0.78** ±0.40 |
| ZH | | | | | | | | | |
| Groud-truth | 4.55 | 3.78 ±0.14 | 0.00 | 4.55 | 3.38 ±0.14 | 0.00 | 4.55 | 3.92 ±0.07 | 0.00 |
| CosyVoice2 | - | - | - | 0.78 | 3.54 ±0.27 | -1.14 ±0.24 | 0.89 | 3.58 ±0.39 | 0.36 ±0.29 |
| VoxInstruct | 3.37 | 3.18 ±0.28 | -1.60 ±0.30 | 10.04 | 2.62 ±0.28 | -1.88 ±0.09 | 9.40 | 3.04 ±0.17 | -1.62 ±0.19 |
| **FlexiVoice-Base** | 5.01 | **4.10** ±0.26 | **0.75** ±0.40 | 3.08 | 3.74 ±0.28 | -1.50 ±0.23 | 5.63 | 3.66 ±0.06 | -0.88 ±0.45 |
| **FlexiVoice** | 7.59 | 4.04 ±0.23 | 0.60 ±0.41 | 4.34 | **3.79** ±0.26 | **-0.36** ±0.24 | 7.02 | **3.76** ±0.29 | **0.57** ±0.44 |

**Multi-modality Control**    Comparing the easy sets of these two tasks, under the guidance of the instruction, TO-easy only includes the text condition (where the model uses a random timbre), while TR-easy incorporates both text and reference speech as multi-modality condition inputs (where the model uses the reference's timbre). Most baselines cannot support both simultaneously, and even those that do, together with FlexiVoice-Base, show a large gap, particularly for Chinese. After progressive post-training, FlexiVoice achieves substantial gains, reaching 97.4% ACC-I in English and 99.8% in Chinese on TO-easy, surpassing ground-truth in some cases. On TR-easy, it maintains strong accuracy (89.4% EN, 81.8% ZH) while preserving speaker consistency, highlighting robust multi-modality control capacity.

**Target Text Disentanglement**    For the two difficulty levels of TO, the easy level uses neutral text while the hard level employs text with emotions differing from the instruction. The purpose is to test whether the model can ignore conflicting emotional cues in text. Baselines and FlexiVoice-Base generally fail here, showing low ACC-I and high ACC-T, indicating they are biased toward the text's implied style. In contrast, FlexiVoice substantially improves disentanglement, achieving 89.4% ACC-I with only 6.6% ACC-T in English, and 98.4% vs. 0.8% in Chinese. Notably, the Chinese gap between easy and hard settings is reduced to just 1.4%, showing FlexiVoice's strong ability to ensure style is controlled by instruction alone.

**Reference Speech Disentanglement**    For TR, the easy task uses neutral speech as the timbre reference, while the hard uses emotionally charged reference speech differing from the instruction. Most baselines exhibit large drops in accuracy and high ACC-R, showing disruption by reference's style. FlexiVoice alleviates this issue, achieving 78.2% ACC-I (EN) and 75.8% (ZH) with correspondingly low ACC-R (10.6% EN, 13.2% ZH). This demonstrates that FlexiVoice effectively disentangles reference timbre from style, preserving timbre while adhering to instruction-defined emotion.

**Trade-off between Style Control and Speaker Verification**    We notice that in certain scenarios (e.g., TR tasks in Table 2), the pre-trained FlexiVoice-Base exhibits slightly higher SV scores compared to the final FlexiVoice. This is an expected phenomenon inherent to the task of style synthesis. The FlexiVoice-Base model, lacking explicit disentanglement capabilities, tends to clone the prosody of the reference speech directly. While this yields high speaker similarity, it fails to follow the style instructions (as evidenced by its low ACC-I scores). In contrast, FlexiVoice must significantly modify acoustic features (such as pitch and energy) to satisfy the target style instruction (e.g., converting a sad reference speech into a happy voice). These necessary prosodic alterations inevitably result in a minor numerical reduction in the cosine similarity of speaker embeddings. Crucially, FlexiVoice maintains high SV scores while achieving superior instruction adherence, demonstrating that it successfully prioritizes style control without compromising the fundamental speaker identity.

**Intelligibility and Subjective Evaluation**    Table 3 presents intelligibility and perceptual quality. FlexiVoice exhibits a marginal increase in WER/CER compared to FlexiVoice-Base. This phe-

Table 4: Complex instruction-following evaluation results on InstructTTSEval, for closed-source and open-source models on both languages.

| Model | InstructTTSEval-EN | | | | InstructTTSEval-ZH | | | |
|---|---|---|---|---|---|---|---|---|
| | APS | DSD | RP | Avg. | APS | DSD | RP | Avg. |
| Groud-truth | 96.2 | 89.4 | 67.2 | 84.3 | 90.9 | 86.7 | 69.8 | 82.5 |
| *Closed-sourced* | | | | | | | | |
| Gemini-flash | 92.3 | 93.8 | 80.1 | 88.7 | 88.2 | 90.9 | 77.3 | 85.4 |
| Gemini-pro | 87.6 | 86.0 | 67.2 | 80.3 | 89.0 | 90.1 | 75.5 | 84.8 |
| GPT-4o-mini-TTS | 76.4 | 74.3 | 54.8 | 68.5 | 54.9 | 52.3 | 46.0 | 51.1 |
| Hume | 83.0 | 75.3 | 54.3 | 71.1 | - | - | - | - |
| *Open-sourced* | | | | | | | | |
| ParlerTTS | 60.0 | 45.9 | 31.2 | 45.7 | - | - | - | - |
| PromptStyle | 57.4 | 46.4 | 30.9 | 38.2 | - | - | - | - |
| VoxInstruct | 54.9 | 57.0 | 39.3 | 50.4 | 47.5 | 52.3 | 42.6 | 47.5 |
| PromptTTS | 64.3 | 47.2 | 31.4 | 47.6 | - | - | - | - |
| MiMo-Audio-7B-Instruct | 80.6 | 77.6 | 59.5 | 72.6 | **75.7** | **74.3** | 61.5 | 70.5 |
| **FlexiVoice-Base** | 63.6 | 75.0 | 60.6 | 66.4 | 56.7 | 59.1 | 59.5 | 58.4 |
| **FlexiVoice** | **81.2** | **85.2** | **71.4** | **79.3** | 71.0 | 71.8 | **69.7** | **70.8** |

nomenon is consistent with prior findings (Li et al., 2023), which indicate that standard ASR models often degrade on highly expressive or emotional speech due to significant prosodic variations. However, this does not imply a loss of intelligibility for human listeners. As evidenced by the consistently higher Q-MOS scores (e.g., 4.08 vs. 3.72 in EN-TO-Easy), FlexiVoice maintains superior perceptual clarity while delivering richer stylistic expression. In addition, the positive CMOS (up to +0.9) indicates better overall expressiveness and instruction adherence, ensuring that the quality of generated speech remains unaffected at the same time. In contrast, baselines often obtain negative CMOS, showing lower preference in human judgments compared with the groud-truth. These results confirm that FlexiVoice not only achieves superior disentanglement but also maintains naturalness and quality close to ground-truth.

## 5.3 COMPLEX INSTRUCTION-FOLLOWING ABILITY

InstructTTSEval (Huang et al., 2025) defines three levels of complex instruction following. These tasks range from low-level acoustic control to open-ended style generation and character imitation, thereby testing different aspects of instruction adherence. As shown in Table 4, the pre-trained model FlexiVoice-Base already performs competitively, especially on DSD (75.0 EN) and RP (60.6 EN), due to alignment between these tasks and the natural instructions present in our constructed dataset. After applying the Progressive Post-Training strategy, FlexiVoice achieves consistent gains across all task types. On English tasks, it improves by over 12 points on APS (81.2 vs. 63.6) and nearly 11 points on RP (71.4 vs. 60.6), reaching an overall average of 79.3, close to Gemini-pro (80.3). On Chinese tasks, FlexiVoice attains 70.8 average accuracy, surpassing MiMo-Audio-7B-Instruct (70.5) and reducing the gap to Gemini models. Overall, FlexiVoice consistently outperforms all open-source baselines and narrows the gap with closed-source commercial systems, demonstrating robust instruction-following ability in complex, real-world scenarios. Furthermore, empirical results in Appendix A.8 confirm that FlexiVoice also excels in fine-grained control of non-emotional attributes like speaking speed and pitch, significantly outperforming baselines in correlation analysis.

## 5.4 EFFECTIVENESS OF PROGRESSIVE POST-TRAINING

To rigorously validate the necessity of the proposed Progressive Post-Training (PPT) scheme, we conduct comprehensive ablation studies comparing different training orders and optimization strategies. The results are summarized in Table 5.

**Importance of Training Order**    We first investigate whether the order of stages matters. As shown in the first block of Table 5, starting directly with complex instruction alignment (+S3) yields poor performance (InstructTTSEval Avg. 72.3 vs. Ours 79.3). This suggests that without the fundamental alignment provided by S1, the model struggles to optimize for the sparse and high-level rewards in S3. Furthermore, the alternative order +S3→S1→S2 results in significantly lower performance on

Table 5: Ablation study of FlexiVoice on different training orders and strategies. We compare the Pre-trained model (Base), the effect of starting with Instruction GRPO (S3 first), Joint Training, and our proposed Progressive Post-Training (PPT) strategy (S1 → S2 → S3).

| Training Strategy | Decoupling Set (EN) | | | | | InstructTTSEval (EN) | | | |
|---|---|---|---|---|---|---|---|---|---|
| | TO Easy | TO Hard | TR Easy | TR Hard | Avg. | APS | DSD | RP | Avg. |
| FlexiVoice-Base | 72.4 | 39.4 | 58.8 | 48.8 | 54.9 | 63.6 | 75.0 | 60.6 | 66.4 |
| *Alternative Training Order* | | | | | | | | | |
| + S3 | 72.8 | 59.0 | 48.6 | 38.2 | 54.7 | 73.3 | 78.8 | 64.7 | 72.3 |
| + S3 → S1 | 96.0 | 79.6 | 80.8 | 72.6 | 82.3 | 74.3 | 81.2 | 67.5 | 74.3 |
| + S3 → S1 → S2 | 94.8 | 80.2 | 86.8 | 75.8 | 84.4 | 74.9 | 81.4 | 68.2 | 74.8 |
| *Joint Training* | | | | | | | | | |
| + S1 → S2 + S3 (Joint) | 94.8 | 80.8 | 86.2 | 74.6 | 84.1 | 78.3 | 80.5 | 67.8 | 75.5 |
| *Progressive Post-Training (PPT)* | | | | | | | | | |
| + S1 | 96.0 | 81.4 | 83.3 | 72.2 | 83.2 | 63.4 | 79.3 | 64.2 | 69.0 |
| + S1 → S2 | 96.4 | 88.0 | 89.4 | **80.2** | 88.5 | 66.8 | 80.3 | 67.9 | 71.7 |
| **+ S1 → S2 → S3 (Ours)** | **97.7** | **89.4** | **89.4** | 78.2 | **88.7** | **81.2** | **85.2** | **71.4** | **79.3** |

complex instructions (74.8) compared to our approach. This indicates that applying the fundamental DPO (S1) after complex instruction training causes catastrophic forgetting of the fine-grained instruction-following capabilities learned in S3. These findings confirm that S1 acts as a necessary "cold start" foundation, establishing robust multi-modal responsiveness before the model tackles more advanced disentanglement and generalization tasks.

**Progressive vs. Joint Training** Another key question is whether S2 and S3 could be optimized simultaneously, since they both apply GRPO algorithm. We compare our progressive approach against a joint training strategy (+S1→S2+S3), where the decoupling and instruction objectives are combined. The results show that joint training underperforms the progressive strategy on both benchmarks (Decoupling Avg. 84.1 vs. 88.7; InstructTTSEval Avg. 75.5 vs. 79.3). We attribute this to the conflicting nature of the gradients: S2 enforces strict constraints to suppress style leakage using fixed classifiers, while S3 encourages open-ended style generalization via ALM-based rewards. Optimizing them jointly leads to interference, preventing the model from mastering either task at the same time.

**Cumulative Gains of Our PPT Strategy** The final block of Table 5 demonstrates the step-by-step efficacy of our proposed path. The Multi-modality DPO (+S1) first yields large gains on basic controllability (+28.3 Decoupling Avg.). Subsequently, the Decoupling GRPO (+S2) significantly boosts the model's ability to separate style from content and timbre, raising the Decoupling Avg. to 88.5. Finally, the Instruction GRPO (+S3) further enhances the model's capability on complex, real-world instructions (+7.6 on InstructTTSEval Avg.) while maintaining the high disentanglement capability achieved in the previous stage. The full PPT curriculum achieves the best balance across all metrics, representing substantial improvements over the base model and validating the progressive design as a stable and effective optimization path.

## 6 CONCLUSION

In this work we introduce FlexiVoice, a TTS system for multi-modality control that uses natural-language instructions for style and reference speech for timbre, supported by a high-quality instruction–speech dataset and a Progressive Post-Training (PPT) paradigm. PPT first strengthens the multi-modality controllability and disentanglement of instruction, text, and reference via emotion-centric DPO/GRPO, then scales to complex instruction following with an ALM-based reward, yielding stable optimization and broad generalization. Experiments show consistent gains on disentanglement (e.g., large ACC-I improvements with low interference from text/reference) and strong performance on InstructTTSEval, where FlexiVoice surpasses competed baselines and narrows the gap to closed-source systems, while maintaining naturalness and robustness in human evaluations.

# 7 ETHICS STATEMENT

This work investigates controllable text-to-speech with multi-modality inputs. The instruction–speech dataset was created from publicly available or licensed resources, filtered to remove offensive, biased, or harmful content. No personal or sensitive user data were collected or processed. While advances in speech synthesis can potentially be misused (e.g., generating deceptive or harmful audio), our study focuses on improving controllability, robustness, and transparency for research purposes, and we encourage responsible use of the proposed methods. All authors have read and adhered to the ICLR Code of Ethics , and confirm that this research complies with standards of fairness, privacy, and research integrity.

# 8 REPRODUCIBILITY STATEMENT

We have taken multiple steps to ensure the reproducibility of our work. Detailed descriptions of the model architecture, training objectives, and evaluation protocols are provided in the main text and appendix. We will release the instruction–speech dataset, model checkpoints, and all training and inference code to facilitate replication and further research. Hyperparameter settings, data processing procedures, and evaluation scripts will also be included in the release materials.

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

## A  APPENDIX

### A.1  MODEL STRUCTURE

As illustrated in Section 3, our model mainly contains two stages: auto-regressive LLM and flow matching. In the first stage, the model receives the inputs of text, instruction, and reference speech. The text and instructions are formatted according to the LLM's input template, with the reference speech transcription concatenated before the text. We use the semantic code extracted from Dual-Codec (Li et al., 2025) to represent the speech in discrete form within our system. Therefore, the reference speech will first be converted into discrete tokens via DualCodec, and the tokens resulting from processing the formatted text and instructions through the LLM encoder will be concatenated to the front. Together, they will serve as input for the auto-regressive LLM. Here we employ Phi-3.5-mini-instruct (Abdin et al., 2024) due to its suitability for multi-modal tasks. We first expand

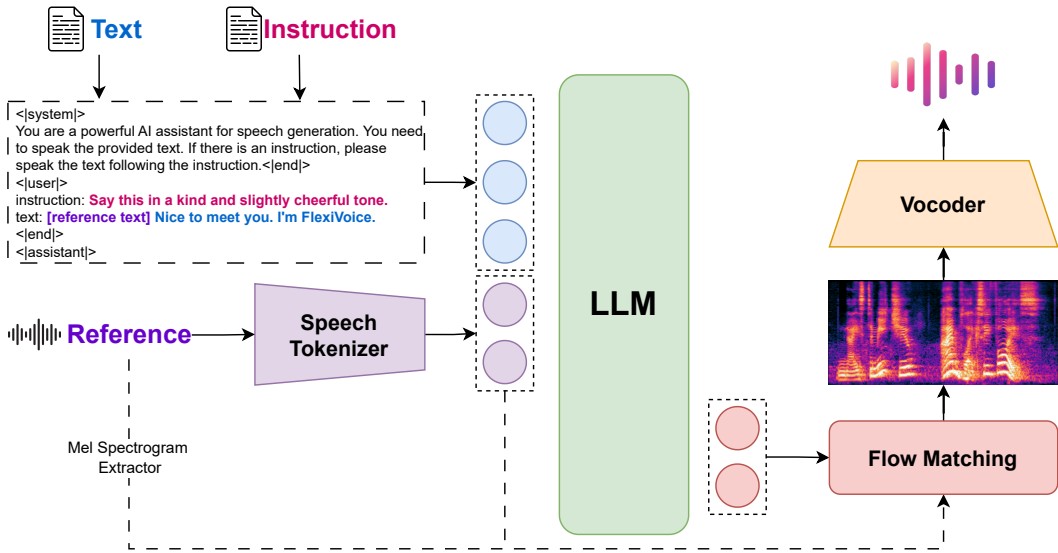

Figure 3: The complete structure of FlexiVoice.

its vocabulary (equivalent to DualCodec's vocabulary size, 16384), then use the parameters of a text LLM as the initial state for pre-training and post-training, illustrated as the main part in our work.

Mainstream TTS works (Du et al., 2024; Zhou et al., 2025; Zhang et al., 2025a) chose flow matching (Lipman et al., 2023) as the decoder because of its high-quality reconstruction of speech details. For the second stage in this work, we employ a flow matching module trained on Emilia (He et al., 2024) to convert generated code into mel-spectrum, using reference speech code as the condition. Finally, the mel-spectrum is converted into the target audio via a vocoder (using Vocos (Siuzdak, 2023) in this case). The model structure is shown in Figure 3

## A.2 DATA PROCESS FOR PRE-TRAINING

Table 6: Instruction-speech datasets used in pre-training stage.

| Source | Dur (h) | Description |
|---|---|---|
| ParaSpeechCaps (Diwan et al., 2025) | 2847 | Open-sourced style-prompted speech data |
| ChildMandarin (Zhou et al., 2024a) | 41 | Speech clips of child's voice |
| Debatts (Huang et al., 2024) | 67 | Speech clips with debating style |
| Emotion set[*] | 117 | Speech clips with different emotions |
| KeSpeech (Tang et al., 2021) | 1541 | Speech clips with different dialects, ages, and genders |
| L2-ARCTIC (Zhao et al., 2018) | 27 | Speech corpus of non-native English |
| NVSpeech (Liao et al., 2025) | 775 | Paralinguistic tagged speech data |
| **FlexiVoice-Instruct** | 4316 | Expressive speech with high-level instructions |

[*] Including CREMA-D (Cao et al., 2014), EMNS (Noriy et al., 2023), ESD (Zhou et al., 2021), IEMOCAP (Busso et al., 2008), M3ED (Zhao et al., 2022), MELD (Poria et al., 2019), MSP-Podcast (Lotfian & Busso, 2019), RAVDESS (Livingstone & Russo, 2018).

For ParaSpeechCaps (Diwan et al., 2025), due to its bottom-up data annotation process, some speech entries possess a detailed acoustic feature dictionary alongside a summarized description. For these data points, we concatenate the feature dictionaries together and randomly sample descriptions. For other data, we utilize their descriptions as instructions.

For ChildMandarin (Zhou et al., 2024a), Detatts (Huang et al., 2024), and the emotion set, since they are all single-label datasets (boy, debate scene, emotion label), we pre-generated several instruction

templates using Deepseek-V3 (Liu et al., 2024a) (e.g., "Speak in the voice of {label}," "Imagine you are in a debate scene," "Express the emotion of {label}"). Then, for each speech data point, we randomly selected a template and filled in the corresponding label.

KeSpeech (Tang et al., 2021) and L2-ARCTIC (Zhao et al., 2018) are both multi-label datasets. For example, each speech in KeSpeech has three attributes: age, gender, and accent. For each attribute triplet across the entire dataset, we use Deepseek-V3 (Liu et al., 2024a) to generate three Chinese instructions and three English instructions, which are then randomly applied to the entire dataset.

For NVSpeech, which is a speech corpus with paralinguistic tags, we do not apply any instruction but use their tags to enlarge the vocabulary size of our core LLM module to guarantee expressive paralinguistic generation. All of the instruction-speech paired datasets in the pre-training stage are listed in Table 6.

## A.3 PROMPTS IN INSTRUCTION DATA CONSTRUCTION

---

### For data source of Emilia

**Role and Tasks**
```
You are a multilingual text analysis expert tasked with generating voice descriptions
required for users' Instruction TTS tasks.  Your core responsibility is:  Generating
voice descriptions based on text and metainfo (including but not limited to scenarios,
styles, emotions, etc.)
```

**Detailed Description**
```
- The metainfo includes the video title and a list of video tags from which the audio
originates.  If the title is irrelevant (e.g., URLs, gibberish, etc.), ignore the
title.
- The scene and speaking style can be inferred from the tags and title, but in case of
conflict, prioritize the text content.
- Output a concise description of the speech in natural human language instructions.
- Use colloquial, vivid phrasing.  Vary sentence structures to avoid template-like
patterns.
- Determine whether valid or rich speech descriptions can be generated based on the
text and meta information and classify the information value (high/medium/low).
```

**Input/Output Specifications**
```
Input Structure:
{
    "speech transcript":  "The speech text to analyze",
    "metainfo":  {"title":  [Title of the source video], "tags":  [Tag list of the
source video]}
}
Output Requirement:
{
    "description":  "Natural language description of the speech style",
    "value":  "information value (high/medium/low)"
}
```

**Example**
```
Example Input:
{
    "speech transcript":  "This match currently shows Ma Chao with four kills, nearly
6,000 in gold.",
    "metainfo":  {"title":  "Honor of Kings KPL Autumn Finals", "tags":
"Game,esports,Honor of Kings,LGD,KPL,DYG"}
}
Example Output:
{
    "description":  "Professional and passionate esports commentary, delivered with
excitement and enthusiasm.",
    "value":  "high"
}
```

**Core Constraints**
```
- When metainfo conflicts with text, prioritize text content.

- Prohibit any unfounded speculation; prohibit all uncertain detail descriptions.

- The example outputs are for reference only; do not rigidly adhere to their phrasing.
```

---

---

**For data source of game voice acting**

**Role and Tasks**
```
You are a multilingual text analysis expert tasked with generating voice descriptions
required for users' Instruction TTS tasks.  Your core responsibility is:  Generating
voice descriptions based on text and metainfo (including but not limited to scenarios,
styles, emotions, etc.)
```

**Detailed Description**
```
- Metainfo refers to a specific character in [GameName], whose dialogue is the speech
transcript.
- Infer the character's speaking style based on their name and output a description of
the speech.
- When the speaking style conflicts with the text's expressed style, prioritize the
text.
- Do NOT include specific character names in the description.
- Use colloquial, vivid phrasing.  Vary sentence structures to avoid template-like
patterns.
- Determine whether valid or rich speech descriptions can be generated based on the
text and meta information and classify the information value (high/medium/low).
```

**Input/Output Specifications**
```
Input Structure:
{
    "speech transcript":  "Speech transcript (dialogue from a [GameName] character)",
    "metainfo":  "Source information (character name in [GameName])"
}
Output Requirement:
{
    "description":  "Natural language description of the speech style",
    "value":  "information value (high/medium/low)"
}
```

**Example**
```
Example Input:
{
    "speech transcript":  "As long as the content is interesting, these experts can
help spice it up.",
    "metainfo":  "Yae Miko"
}
Example Output:
{
    "description":  "With a hint of playful elegance in her tone, as if she had long
seen through the other's thoughts yet chose to hint at them subtly.",
    "value":  "high"
}
```

**Core Constraints**
```
- When metainfo conflicts with text, prioritize text content.

- Prohibit any unfounded speculation; prohibit all uncertain detail descriptions.

- The example outputs are for reference only; do not rigidly adhere to their phrasing.
```

## A.4   DATA CONSTRUCTION FOR GRPO TRAINING

**Decoupling Task**   The training set is constructed from the recording subset of NCSSD (Liu et al., 2024b), comprising about 20,000 Chinese and 10,000 English dialogue speech samples. We selected this dataset primarily because its transcriptions align closely with everyday conversations and implicitly carry emotional tendencies in most cases. We randomly assign emotion labels to each data point, thereby generating two types of data: those where the instruction and transcription emotions align, and those where they conflict. This approach helps guide the model to learn the ability to decouple content and style within text. Simultaneously, we randomly assign neutral and emotional speech from the pre-training data's emotion set (6) to each data point with a 90% and 10% probability, respectively, as reference speech. This enables the model to learn to decouple timbre and style from the references. We utilize half of the Chinese data and all of the English data, forming approximately 20,000 balanced data points as the GRPO training set for the decoupling phase.

**Complex Instruction**   To enhance the model's performance in complex instruction-following tasks, we constructed a rich and diverse GRPO training dataset. Specifically, for each language (Chinese and English), we first randomly sampled 1,000 existing instruction-text inputs from the

pre-training data to ensure optimization stability. Next, we prompted Deepseek-V3 (Liu et al., 2024a) to randomly generate 6,000 instructions across three distinct configurations: (1) Generate detailed and comprehensive acoustic feature descriptions in dictionary format, (2) Generate natural language descriptions incorporating 3/4/5/6 explicit acoustic features, (3) Freely generate instructions for arbitrary scenarios consistent with human usage patterns. Collectively, we generated 14,000 instruction-text paired inputs as GRPO training data for the second stage.

## A.5    DETAILS IN EMOTIONAL CONTROL DPO

> **Instruction Templates**
>
> - Say the sentence with the emotion of {label}.
> - Say this with a {label} tone.
> - Speak this sentence in a {label} manner.
> - Deliver this text with {label} emotion.
> - Use a {label} voice.
> - Express this sentence with {label} feeling.
> - Read it with {label} inflection.
> - Recite this with {label} sentiment.
> - Voice this in a {label} style.
> - {label} (Use only one emotion label)
>
> - 用{label}的情感说出这句话。
> - 以{label}的情绪表达这句话。
> - 带着{label}的情感说出这个文本。
> - 使用{label}的情感。
> - 以{label}的心情表达这句话。
> - 用{label}的情感色彩说出这句文本。
> - 带着{label}的感情说出这句话。
> - 用{label}的情感基调传达此句。
> - 以{label}的情绪状态说出这句话。
> - {label}

> **One Paired Sample (emo=angry, speaker_id=0013)**
>
> {
>
>     "prompt_text": "Chapter eleven on the doorstep.",
>     "prompt_wav_path": "Emotion_Speech_Dataset**/0013/**Neutral/0013_000267.wav",
>     "target_text": "What are you waiting for? man.",
>     "instruction": "Say this with a angry tone.",
>     "chosen_wav_path": "Emotion_Speech_Dataset**/0013/**Angry/0013_000697.wav"
>     "rejected_wav_path": "Emotion_Speech_Dataset**/0013/**Sad/0013_001397.wav"
>
> }

## A.6    REWARD SELECTION IN DECOUPLING GRPO

In decoupling GRPO, we first use a SER signal as one of the rewards to control the correct emotion synthesis. With the reference speech input, we also need to guarantee that the generated result is from the same speaker with the reference. There are two choices: speaker verification signal (0 or 1) and speaker similarity signal (values between 0 and 1), using CAM++ (Wang et al., 2023) and WavLM-large-finetuned (Chen et al., 2022) respectively. Given the DPO optimized model, we train the multi-objective GRPO using SER reward with two separate speaker-related rewards for four epochs, then test them on two difficulty levels of TR in the decoupling evaluation set, as shown in Figure 4.

Results show that models using speaker verification as the reward signal achieve significant improvements, while those using speaker similarity yield largely unchanged or even reduced performance. This occurs because models computing speaker embeddings (WavLM in this case) do not rely solely on speech timbre but incorporate additional acoustic features beyond timbre, such as pitch and emotion. When humans speak with varying emotions, acoustic attributes like pitch, speech rate, and volume inevitably differ. Consequently, expressive emotion delivery yields high SER reward scores but low speaker similarity signals. This phenomenon can thus be interpreted as a conflict between

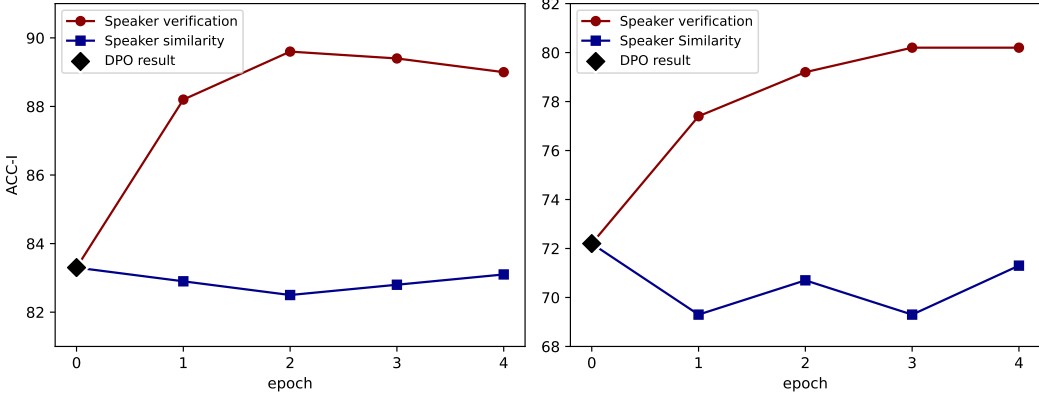

Figure 4: Comparison of results for two reward signals in decoupling GRPO, on the decoupling evaluation set's TR-easy (left) and TR-hard (right) tasks.

SER reward and speaker similarity reward, hindering the normal optimization process of the multi-objective GRPO task.

Speaker verification signals, as a more loose form of speaker similarity, can accommodate models that generate more expressive emotional speech while ensuring speaker identity. Therefore, we select speaker verification as one of the final reward signals in the S1 GRPO stage.

### A.7 Validation of Reward Models

For the complex instruction following in Stage 3 (S3), we require a robust reward model to evaluate open-ended generation. While Gemini-2.5-Pro serves as the Gold Standard judge in the InstructTT-SEval benchmark and aligns well with human preference (Huang et al., 2025), employing it directly as a reward model for GRPO is impractical due to its prohibitive inference costs and high latency during online sampling. Consequently, we propose utilizing Kimi-Audio-7B-Instruct as a cost-effective surrogate.

To validate the reliability of this substitution, we assess the alignment between Kimi-Audio-7B and Gemini-2.5-Pro. We conduct the evaluation on the InstructTTSEval dataset, measuring the agreement of their judgments using the Macro-F1 score under the same prompt used in our training. As shown in Table 7, Kimi-Audio-7B achieves solid agreement (Macro-F1 $> 0.60$) with Gemini across all task types in both English and Chinese. These results demonstrate that Kimi-Audio-7B effectively captures the preference patterns of the larger model, providing valid and efficient signals for optimizing complex instruction adherence.

Table 7: Agreement evaluation (Macro-F1) between our surrogate reward model (Kimi-Audio-7B) and the gold-standard judge (Gemini-2.5-Pro) on InstructTTSEval. The high consistency validates the effectiveness of using Kimi-Audio as a reward model.

| Metric | English (EN) | | | Chinese (ZH) | | |
|---|---|---|---|---|---|---|
| | APS | DSD | RP | APS | DSD | RP |
| Macro-F1 | 0.62 | 0.62 | 0.65 | 0.60 | 0.64 | 0.63 |

### A.8 Evaluation on Non-Emotional Style Control

While our main experiments focus on emotional expression and complex instruction following, the claim of controlling flexible style necessitates validation on fine-grained prosodic attributes beyond emotion. To address this, we conducted specific evaluations on **Speaking Speed** and **Pitch** control.

Table 8: Spearman Correlation Coefficient evaluation on Speaking Speed and Pitch control.

| Model | Speed Correlation | | Pitch Correlation | |
|---|---|---|---|---|
| | EN | ZH | EN | ZH |
| ParlerTTS | 0.61 | 0.22 | 0.68 | 0.79 |
| PromptTTS | 0.24 | - | 0.12 | - |
| PromptStyle | 0.05 | - | 0.24 | - |
| VoxInstruct | 0.75 | 0.49 | 0.31 | 0.15 |
| FlexiVoice-Base | 0.62 | 0.47 | 0.66 | 0.52 |
| **FlexiVoice** | **0.86** | **0.78** | **0.91** | **0.90** |

We constructed a test set comprising 100 samples for English and 100 for Chinese. For each sample, we synthesized speech conditioned on instructions corresponding to three distinct levels:

- **Speed:** *Fast*, *Normal*, *Slow* (e.g., instruction: Read this sentence quickly.)
- **Pitch:** *High*, *Normal*, *Low* (e.g., instruction: Speak with a high-pitched voice.)

To quantify the controllability, we calculated the **Spearman Correlation Coefficient** between the instruction levels (mapped to 1, 2, 3) and the extracted acoustic features of the generated audio. Specifically, we measured phoneme duration (phonemes per second) for speed and median F0 for pitch. A higher correlation indicates that the model more accurately follows the gradient of the instruction.

As shown in Table 8, FlexiVoice achieves the highest correlation scores across both languages and attributes. Notably, it significantly outperforms the baseline models and the pre-trained FlexiVoice-Base, demonstrating that our Progressive Post-Training (PPT) strategy effectively enhances fine-grained physical parameter control alongside abstract emotional styles, even we do not apply any targeted optimization on speed or pitch.

## A.9 SUBJECTIVE EVALUATION CONFIGURATION

Subjective evaluations are carried out by a group of compensated participants, all of whom had strong speech and audio expertise. We selected two baseline models with relatively comprehensive task support, our pre-trained model, and FlexiVoice for subjective evaluation. For each task in each language, we randomly select five paired samples from each model and the ground-truth. To ensure reliability, every audio sample was independently rated by at least three different individuals for both subjective evaluation.

**Q-MOS**    Participants are asked to evaluate the overall quality of each generated speech sample on a 5-point scale, considering aspects such as clarity, naturalness, and absence of distortion/artifacts, ignoring the style instruction and timbre reference. The meaning of each score is defined as follows:

- **5** - Speech is highly natural, clear, and pleasant to listen to. No noticeable artifacts or distortions. Comparable to professionally recorded human speech.
- **4** – Speech is generally natural and intelligible, with only minor imperfections or occasional artifacts that do not interfere with understanding or listening comfort.
- **3** – Speech is intelligible but has moderate issues such as slight distortion, unnatural prosody, or mild background noise. Quality is acceptable but clearly below high-standard human recordings.
- **2** – Speech is somewhat difficult to understand due to significant artifacts, distortions, or unnatural delivery. Quality issues noticeably affect the listening experience.
- **1** – Speech is largely unintelligible or highly unnatural, with severe artifacts or distortions that make evaluation difficult.

**CMOS**   In this task, participants are asked to compare two audio samples (one ground-truth and one generated by a model) under the condition of a given target emotion. The primary focus is on the richness and naturalness of emotional expression, which reflects the instruction-following and decoupling ability of the models.

The additional evaluation rules are: (1) When no reference speech is provided, ignore timbre consistency. (2) When reference speech is provided, timbre similarity is a secondary criterion (acceptable as long as both samples sound like the same speaker). (3) Slight mispronunciations or noise should be disregarded; the main comparison is the accuracy and expressiveness of emotion. And the scoring scale is defined as follows: (Note that the order of paired audio demonstrations is random.)

- **+2** - Audio B is much better than Audio A in conveying the target emotion.
- **+1** – Audio B is slightly better than Audio A in emotional expression.
- **0** – Both samples are comparable in terms of emotional richness and naturalness.
- **−1** – Audio A is slightly worse than Audio B in emotional expression.
- **−2** – Audio A is much worse than Audio B in emotional expression.

## A.10   REPRODUCIBILITY DETAILS

To facilitate the replication of our results and future research, we provide detailed specifications regarding the computational resources and training configurations for the Progressive Post-Training (PPT) scheme. FlexiVoice is built upon the Phi-3.5-mini-instruct backbone, which comprises approximately 3.8 billion parameters. All training stages are conducted on $8\times$ NVIDIA A800 (80GB) GPUs. The total training time for the post-training pipeline is approximately 3.5 days.

Specifically, the S1 (Multi-modality DPO) stage serves as an efficient alignment foundation, completing in approximately 2 hours. We train the model for 3 epochs with a learning rate of $1 \times 10^{-5}$ and a KL penalty coefficient $\beta = 0.1$. The S2 (Decoupling GRPO) stage, which involves online sampling to enforce disentanglement, requires approximately 36 hours. For this stage, we employ a group size of $G = 8$, training for 2 epochs with a learning rate of $1 \times 10^{-5}$ and $\beta = 0.1$. Finally, the S3 (Instruction GRPO) stage utilizes a group size of $G = 6$ to accommodate complex instruction following and runs for 2 epochs. Due to the complexity of the ALM-based reward calculation, this stage takes approximately 42 hours.

## A.11   LLM USAGE

Large Language Models (LLMs) were used solely as an assistive tool for grammar correction and language polishing of the manuscript. They did not contribute to research ideation, methodology, experiments, analyses, or the generation of scientific content. All conceptual and technical contributions are entirely those of the authors.

