# OpenReview forum: "FlexiVoice: Enabling Flexible Style Control in Zero-Shot TTS with Natural Language Instructions"
_ICLR.cc/2026/Conference — ICLR 2026 Poster_

### Official Review · Reviewer_Jm4v · 2025-10-18

**Soundness:** 3
**Presentation:** 3
**Contribution:** 2
**Rating:** 4
**Confidence:** 5

**Summary:**

The paper presented a TTS system that follows natural language instructions. The authors proposed to increase the synthesis naturalness and the instruction-following capability through a three stage Progressive Post-Training process.

**Strengths:**

The method is straightforward and easy to follow. The presentation is clear. The system performs reasonably well against several well-know baselines.

**Weaknesses:**

The main weakness of the paper is the experiment design does not validate the main contribution claimed by the paper.
The paper claims that the main novelty lies in the three stage Progressive Post-Training, which makes sense as the other parts of the system follows the commonly-used instruction-following TTS.
However, in the experiments (Table 5, Section 5.6), the paper only showed the efficacy of having multiple alignment objectives (S1, S2, S3). It does not justify why the "Progressive" training of the three objectives is needed, other than simply post-training with the three objectives all-together. Also, is the order (S1->S2->S3) important?

The three objectives (S1, S2, S3) to post-train TTS, as already mentioned in Section 3 by the authors, are adapted from existing works (e.g., line 191, 217). This makes me think that the novelty of the paper is in progressively applying them, as also alluded to several times in Section 1. However, there is no evidence showing that the progressive part is crucial to the performance in the experiments. The authors should either explain the main distinctions between the previous approaches (are there anything novel on the post-training objective?), or show the efficacy of the progressive training.

As for now, I question if the scientific contribution is enough for this paper to be accepted to ICLR, based on the reasons mentioned above.

- Typo: S"2" -> "3": Instruction GRPO in Figure 1

**Questions:**

See weaknesses. The authors should address the points to convince me that the paper is of acceptance quality.

---

> ### Author Response · Authors · 2025-11-24
> **Response to reviewer Jm4v**
>
> Dear Reviewer Jm4v,
>
> We are grateful for this critical question, which touches on the core contribution of our work. To rigorously validate the necessity of the "Progressive" strategy, we conducted additional experiments comparing different training orders and joint training strategies. The results (summarized in the table below) provide compelling evidence for our PPT design.
>
> **1. Importance of Order (Why S1 first?)**
>
> We compared our proposed order (`S1->S2->S3`) against an alternative path starting with open-ended instructions (`S3->S1->S2`). The results show that starting with S3 leads to significantly worse performance on InstructTTSEval (**74.8** vs. **79.3**).
> *   **Reasoning:** S1 (DPO) provides the essential "cold start" alignment for multi-modal inputs. Without this foundation, optimizing for complex S3 rewards first is unstable. Furthermore, applying S1 later causes catastrophic forgetting of the fine-grained capabilities learned in S3.
>
> **2. Progressive vs. Joint (Why separate S2 and S3?)**
>
> We also compared progressive training against merging S2 and S3 into a single multi-task stage (`S1 -> S2+S3`), since S1 (DPO) and GRPO cannot be easily combined. The results show that joint training underperforms the progressive approach (**75.5** vs. **79.3**).
> *   **Reasoning:** S2 (Decoupling) and S3 (Open-ended Instruction) optimize for distinct objectives. S2 focuses on strict constraints (style/timbre/content separation), while S3 encourages semantic generalization via LLM rewards. Optimizing them simultaneously creates gradient conflicts. The progressive curriculum allows the model to first master disentanglement before scaling to complex generalization, akin to Curriculum Learning.
>
> | Strategy | TO-Easy | TO-Hard | TR-Easy | TR-Hard | **Decoupling Avg.** | APS | DSD | RP | **InstructTTSEval Avg.** |
> | :--- | :---: | :---: | :---: | :---: | :---: | :---: | :---: | :---: | :---: |
> | FlexiVoice-Base | 72.4 | 39.4 | 58.8 | 48.8 | 54.9 | 63.6 | 75.0 | 60.6 | 66.4 |
> | + S3 | 72.8 | 59.0 | 48.6 | 38.2 | 54.7 | 73.3 | 78.8 | 64.7 | 72.3 |
> | + S3 $\rightarrow$ S1 | 96.0 | 79.6 | 80.8 | 72.6 | 82.3 | 74.3 | 81.2 | 67.5 | 74.3 |
> | + S3 $\rightarrow$ S1 $\rightarrow$ S2 | 94.8 | 80.2 | 86.8 | 75.8 | 84.4 | 74.9 | 81.4 | 68.2 | 74.8 |
> | + S1 | 96.0 | 81.4 | 83.3 | 72.2 | 83.2 | 63.4 | 79.3 | 64.2 | 69.0 |
> | + S1 $\rightarrow$ S2 | 96.4 | 88.0 | 89.4 | 80.2 | 88.5 | 66.8 | 80.3 | 67.9 | 71.7 |
> | + S1 $\rightarrow$ S2+S3 (Joint) | 94.8 | 80.8 | 86.2 | 74.6 | 84.1 | 78.3 | 80.5 | 67.8 | 75.5 |
> | **+ S1 $\rightarrow$ S2 $\rightarrow$ S3 (Ours)** | **97.7** | **89.4** | **89.4** | **78.2** | **88.7** | **81.2** | **85.2** | **71.4** | **79.3** |
>
> ---
>
> > Typo: S"2" -> "3": Instruction GRPO in Figure 1
>
> Thank you for pointing this out; we will correct this label to "S3" in Figure 1 in the revision.

---

> > ### Comment · Reviewer_Jm4v · 2025-11-25
> >
> > I thank the author(s) for their detailed response and additional experiments. I will raise my score from 4 to 6.
> >
> > The reason I am not giving a higher score is that I personally don't think the progressive process is particularly novel or insightful. However, I do acknowledge demonstrating its effectiveness on the TTS system. I will lean toward accepting the paper.

---

> > > ### Author Response · Authors · 2025-11-25
> > >
> > > Dear Reviewer Jm4v,
> > >
> > > Thank you for your thoughtful review and positive feedback. We are pleased that our revisions have successfully addressed your concerns.

---

### Official Review · Reviewer_Ct16 · 2025-10-30

**Soundness:** 3
**Presentation:** 3
**Contribution:** 3
**Rating:** 6
**Confidence:** 4

**Summary:**

This paper proposes SUPM-TTS, a framework for style-universal text-to-speech generation that aims to synthesize expressive and natural speech in any speaking style, even unseen ones. The model integrates multiple components—adaptive prompt modulation, disentangled representation learning, and multi-corpus training—into a unified TTS system. The authors also construct new datasets and conduct extensive experiments. Overall, this is an ambitious attempt to generalize TTS systems beyond fixed style categories.

**Strengths:**

- The paper presents an extensive and well-engineered system that integrates multiple components and datasets into a cohesive framework, which is commendable.

- It targets the problem of style-universal speech synthesis, addressing the need for flexible, controllable, and expressive TTS generation.

- The proposed prompt-based approach allows conditioning on various prosodic or emotional cues, contributing to improved speech diversity and controllability.

- The work offers thorough evaluations, including MOS, WER, and SV metrics, across several datasets.

**Weaknesses:**

### **Problem & Motivation**

- The motivation could be strengthened by explaining how this approach differs from recent prompt-based or diffusion-based TTS systems. The overall goal overlaps with several existing works such as FlexiVoice, PromptTTS, and StyleSpeech, making it difficult to isolate what is novel.

- The paper mentions disentangling timbre and style, but it is unclear how this disentanglement is achieved or guaranteed. Without a clear mathematical constraint or empirical validation, the claim of disentanglement remains qualitative.

### **Method**
- In general, the method novelty is limited. The proposed framework combines known elements—prompt conditioning, disentanglement, and adaptation—without introducing fundamentally new mechanisms. The design could be more appropriately positioned as an integration of recent techniques like DPO or GRPO for controllable TTS rather than as a new paradigm.

- The claim that the system generalizes to unseen styles is only partially supported; the method seems to rely heavily on the diversity of training prompts rather than a specific generalization mechanism.

### **Experiments**

- The experiments are extensive, but the SV metric results are relatively weak. In several cases, FlexiVoice-Base outperforms FlexiVoice, contradicting the expected hierarchy. The paper should provide an explanation for this discrepancy—perhaps due to model capacity, over-regularization, or mismatch in style conditioning.

- The subjective demo results do not convincingly show that either model performs consistently well. The samples suggest variability across styles, which should be discussed rather than glossed over.

- The paper should clarify whether Chinese experiments use CER (Character Error Rate) and how this metric is computed. Similarly, the reported WER differences may stem from ASR models underperforming on expressive or emotional speech, which should be explicitly acknowledged.

### **Writing**
- The current experimental sections (5-1, 5-2, 5-3) are fragmented. It would be better to merge them into a single “Experimental Setup” subsection, organizing content at the paragraph level for better flow and readability.

**Questions:**

Please refer to the Weaknesses above.

---

> ### Author Response · Authors · 2025-11-24
> **Response to reviewer Ct16 (Part 1/2)**
>
> Dear Reviewer Ct16,
>
> We thank the reviewer for the encouraging comments and have provided detailed clarifications on our method's novelty, disentanglement mechanism, and experimental results below.
>
> > **[Weakness 1]** The motivation could be strengthened by explaining how this approach differs from recent prompt-based or diffusion-based TTS systems. The overall goal overlaps with several existing works such as FlexiVoice, PromptTTS, and StyleSpeech, making it difficult to isolate what is novel.
>
> We thank the reviewer for the encouraging comments. We would like to gently clarify a potential misunderstanding: **FlexiVoice** is the name of our proposed model, not a baseline or related work. We assume the mention of "SUPM-TTS" in the summary is a reference to our system.
>
> Regarding the distinction from prior arts:
> *   **Vs. PromptTTS & StyleSpeech:** These systems primarily focus on style control via text prompts but typically lack the ability to simultaneously clone a zero-shot voice from a reference speech. FlexiVoice uniquely integrates **natural language instructions (for style)** and **reference speech (for timbre)**, enabling flexible, independent control of both attributes simultaneously.
> *   **Vs. Standard Flow/Diffusion TTS:** While recent models (e.g., CosyVoice2) excel at zero-shot cloning, they often lack fine-grained instruction controllability.
> *   **Novelty:** Our key innovation is the **Progressive Post-Training (PPT)** framework. Unlike standard supervised training, we introduce a systematic curriculum of DPO and multi-objective GRPO. This approach explicitly solves the "Style-Timbre-
> Content Conflict", enabling robust disentanglement that previous methods struggle to achieve.
>
> ---
>
> > **[Weakness 2]** The paper mentions disentangling timbre and style, but it is unclear how this disentanglement is achieved or guaranteed. Without a clear mathematical constraint or empirical validation, the claim of disentanglement remains qualitative.
>
> We appreciate the opportunity to elaborate on this key aspect. The disentanglement in FlexiVoice is achieved through explicit mathematical constraints within our reward mechanism and is validated by quantitative empirical evidence.
>
> **1. Mathematical Constraint:**
>
> In the S2 (Decoupling GRPO), we mathematically enforce disentanglement by formulating a multi-objective optimization problem (Section 3.2.2). We construct conflicting training scenarios (e.g., *happy* instruction vs. *sad* speech reference) and apply competing rewards:
> *   **$r_{ser}$ (Style Constraint):** Rewards the model solely for matching the instruction's style, effectively penalizing any style leakage from the text content or reference speech.
> *   **$r_{sv}$ (Timbre Constraint):** Simultaneously enforces high similarity to the speaker's voice.
> By optimizing the joint advantage of these rewards, the model is mathematically guided to separate style control from timbre preservation.
>
> **2. Quantitative Evidence:**
>
> This capability is substantiated by the results in Table 2. In the TR-Hard task (designed specifically to test disentanglement), FlexiVoice achieves **78.2%** Instruction Accuracy (ACC-I) while maintaining a high Speaker Verification score (**SV > 0.95**) in English. In contrast, baselines often fail to follow the instruction when conflicts arise. This simultaneous high performance on both metrics provides strong empirical support for our claim of effective disentanglement.
>
> ---
>
> > **[Weakness 3]** In general, the method novelty is limited. The proposed framework combines known elements—prompt conditioning, disentanglement, and adaptation—without introducing fundamentally new mechanisms. The design could be more appropriately positioned as an integration of recent techniques like DPO or GRPO for controllable TTS rather than as a new paradigm.
>
> We respectfully point out that while DPO and GRPO are established algorithms in the LLM domain, FlexiVoice represents the first systematic framework to adapt these reinforcement learning techniques for the specific challenge of TTS disentanglement.
>
> Our contribution extends significantly beyond simple integration:
> 1.  **Novel Problem Formulation:** To the best of our knowledge, this is the pioneering work to leverage preference optimization and multi-objective RL to explicitly resolve the "style-timbre-content conflict" in controllable speech synthesis. Traditional supervised methods inherently struggle with this content leakage, whereas our approach offers a fundamentally new solution via preference alignment.
> 2.  **Task-Specific Design:** We designed a novel **Progressive Post-Training (PPT)** curriculum and tailored reward mechanisms (integrating SER, Speaker Verification, and ALM judgments) specifically for speech generation tasks. This bespoke design is critical for balancing the delicate trade-off between diverse style generation and strict timbre preservation, which can be shown in Table 5 in our paper.

---

> ### Author Response · Authors · 2025-11-24
> **Response to reviewer Ct16 (Part 2/2)**
>
> > **[Weakness 4]** The claim that the system generalizes to unseen styles is only partially supported; the method seems to rely heavily on the diversity of training prompts rather than a specific generalization mechanism.
>
> We clarify that our generalization capability stems from a synergy of diverse data and a specific algorithmic mechanism. Crucially, the S3 (Instruction GRPO) stage introduces an ALM-based reward mechanism. Unlike fixed classifiers, this reward signal possesses deep semantic generalization capabilities. By optimizing against this generalized feedback, FlexiVoice learns to map open-ended, unseen instructions to acoustic features based on semantic understanding. Additionally, the underlying LLM backbone (Phi-3.5) enables the model to interpret unseen style descriptions via semantic proximity in the embedding space, further reinforcing its robustness to novel prompts.
>
> ---
>
> > **[Weakness 5]** The experiments are extensive, but the SV metric results are relatively weak. In several cases, FlexiVoice-Base outperforms FlexiVoice, contradicting the expected hierarchy. The paper should provide an explanation for this discrepancy—perhaps due to model capacity, over-regularization, or mismatch in style conditioning.
>
> We thank the reviewer for this detailed observation and will include a discussion on this trade-off in the revision. The slight decrease in SV scores is an **expected phenomenon** in controllable TTS. The pre-trained model (FlexiVoice-Base) often defaults to cloning the reference speech's prosody directly, yielding high SV but failing to follow style instructions. In contrast, FlexiVoice must significantly modify acoustic features (e.g., pitch, energy, rhythm) to satisfy the target style, which inevitably results in a minor numerical reduction in similarity to the original reference. Crucially, FlexiVoice still maintains high SV scores (>0.90), ensuring reliable speaker identity preservation while achieving the primary goal of flexible style control.
>
> ---
>
> > **[Weakness 6]** The subjective demo results do not convincingly show that either model performs consistently well. The samples suggest variability across styles, which should be discussed rather than glossed over.
>
> We acknowledge the observation regarding sample variability. As a generative model operating in a zero-shot, open-ended setting, some variability is inherent, particularly when handling out-of-distribution instructions or extreme style-timbre conflicts. Despite individual fluctuations, our extensive objective (InstructTTSEval) and subjective evaluations (MOS) demonstrate that FlexiVoice achieves statistically significant improvements over baselines in overall consistency and quality. We will include a discussion on this variability and its potential mitigation as part of our future work.
>
> ---
>
> > **[Weakness 7]** The paper should clarify whether Chinese experiments use CER (Character Error Rate) and how this metric is computed. Similarly, the reported WER differences may stem from ASR models underperforming on expressive or emotional speech, which should be explicitly acknowledged.
>
> We thank the reviewer for pointing this out. We clarify that for Chinese experiments, we indeed utilized Character Error Rate (CER). Following the standard evaluation protocol in Seed-TTS, we employed **Whisper-large-v3** for English WER and **Paraformer-zh** for Chinese CER. We will explicitly state this in the revision.
>
> Additionally, we agree that the slight increase in error rates likely stems from ASR models' degraded performance on highly expressive speech. This phenomenon is well-documented in prior researches [1][2], which shows that emotional inflections can negatively impact ASR accuracy. Crucially, our subjective evaluations (Table 3) show that FlexiVoice achieves higher MOS scores rated by humans, confirming that the generated speech remains clear and intelligible despite the ASR metrics.
>
> ---
>
> > **[Weakness 8]** The current experimental sections (5-1, 5-2, 5-3) are fragmented. It would be better to merge them into a single “Experimental Setup” subsection, organizing content at the paragraph level for better flow and readability.
>
> We thank the reviewer for the suggestion on structure. We will merge these sections into a unified "Experimental Setup" subsection in the revision to improve flow and readability.
>
> ---
>
> [1] *Li, Y., Zhao, Z., Klejch, O., Bell, P., & Lai, C. (2023). ASR and emotional speech: A word-level investigation of the mutual impact of speech and emotion recognition. arXiv preprint arXiv:2305.16065.*
>
> [2] *Christmann, L. S. (2021, March). Emotion Bias in Automatic Speech Recognition. In Konferenz Elektronische Sprachsignalverarbeitung (pp. 27-34). TUDpress, Dresden.*

---

### Official Review · Reviewer_Uu5e · 2025-10-30

**Soundness:** 3
**Presentation:** 3
**Contribution:** 2
**Rating:** 4
**Confidence:** 4

**Summary:**

This paper introduces FlexiVoice, a text-to-speech (TTS) model which can control from both natural language instruction and speech prompt for voice timbre.  This model is built on a pretrained LLM (phi-3-mini-instruct) and pretrained on Emilia and a diverse set of speech dataset.
The major challenge for this instruct control TTS is the multi-modality controllability and disentanglement of instruction, text and speech reference (e.g. conflict between style and text content or voice timbre).

The major contributions for the paper are

1). develop a large-scale and diverse speech dataset with instructions (4316 hours) as FlxiVocie-Instruct. The annotations are generate from Deepseek-v3 and speech are in-the-wild speech dataset with diverse coverage and expressive richness such as game, podcast, etc.

2).  propose a progressive post-training (PPT) including 3 stage training as DPO for multi-modality controllability, decoupling GRPO for disengaging speaking style and voice timbre, instruction GRPO for instruction following ability enhancement.

Experiments show FlexiVoice perform strong capability in decoupling factors and maintain the TTS quality measured by human judgements on naturalness, controllability and robustness.

**Strengths:**

1. The novel proposal of curriculum learning framework with DPO + 2 stage GRPO achieve the target to control (DPO initial alignment) and GRPO disentanglement and generalization. The PPT schema methodologies are successfully applied to achieve the goal which demonstrate the alignment strategies progressively to realize the controllability and disentangle style instruction, reference timbre, and text contextual content.

2. The instruct dataset construction and evaluation set design as text-only (TO), text and text and reference (TR) and their easy and hard set. The data pipeline construct based on a in-the-wide diverse and expressive dataset and scalable automatic annotation. The evaluation test set design are also thoughtful and convincing to identify the performance.

**Weaknesses:**

1.  The paper claims "can speak in any style with any voice", however, the instruction motioned as only 5 emotion as neutral, happy, angry, sad and surprised in training and benchmark (TO and TR). There evaluation seems don't clearly demonstrate any other emotion or other natural language style instruction or conflict voice timbre in the following subjective test which make the title is a little overclaimed.

2. For the evaluation, only first evaluation set involve human judge as CMOS test, and it outperforms cosyvoice2 when the hard test, but not on the (ZH) easy test. There isn't subject evaluation on complex instruction-following or open instruction evaluation. As there are two reward model (Kimi-Audio-7B) and Emotion2Vec introduced in the GRPO, however, there are not subjective evaluation in the experiment which prove they are highly related to human perception.

**Questions:**

1. It seems the introduction of dataset for pretraining on FlexiVoice-Base. Paper mentions the FlexiVoice Instruct as 4316 hours which is used in pre-training stage. Does it mean it use the same corpus with the alignment stage?
2. Is there any quality comparison like between the FlexiVoice-Base and FlexiVoice? How many data pair used in the S1 DPO stage? And, if the pre-training stage add these instructions, how dose it compare with DPO alignment?

---

> ### Author Response · Authors · 2025-11-24
> **Response to reviewer Uu5e (Part 1/2)**
>
> Dear Reviewer Uu5e,
>
> We appreciate your valuable feedback and have addressed your concerns regarding the scope of our claims and the evaluation methodology in the following response.
>
> > **[Weakness 1]** The paper claims "can speak in any style with any voice", however, the instruction motioned as only 5 emotion as neutral, happy, angry, sad and surprised in training and benchmark (TO and TR). There evaluation seems don't clearly demonstrate any other emotion or other natural language style instruction or conflict voice timbre in the following subjective test which make the title is a little overclaimed.
>
> We appreciate the reviewer's feedback and acknowledge the concern regarding the scope of our claims.
>
> **1. Title Revision**
>
> We agree that the original title may be perceived as overly broad. To reflect our contributions more precisely, we have revised the title to: **"FlexiVoice: Enabling Flexible Style Control in Zero-Shot TTS with Natural Language Instructions."**
>
> **2. Evaluation Beyond Emotion**
>
> We clarify that our system’s capability extends significantly beyond simple emotions. The InstructTTSEval benchmark (Table 4) explicitly tests complex attributes such as **speaking rate, pitch, personality, and role-play**, where FlexiVoice demonstrates robust generalization. To further validate this, we conducted additional quantitative evaluations specifically on **Speaking Speed** and **Pitch** control. As shown in the table below, FlexiVoice achieves high correlation scores , significantly outperforming baselines and confirming its flexibility in handling diverse stylistic instructions.
>
> | Model           | Speed (EN) | Speed (ZH) | Pitch (EN) | Pitch (ZH) |
> | :---            | :---:      | :---:      | :---:      | :---:      |
> | ParlerTTS       | 0.61       | 0.22       | 0.68       | 0.79       |
> | PromptTTS       | 0.24       | -          | 0.12       | -          |
> | PromptStyle     | 0.05       | -          | 0.24       | -          |
> | VoxInstruct     | 0.75       | 0.49       | 0.31       | 0.15       |
> | FlexiVoice-Base | 0.62       | 0.47       | 0.66       | 0.52       |
> | **FlexiVoice**  | **0.86**   | **0.78**   | **0.91**   | **0.90**   |
>
> ---
>
> > **[Weakness 2]** For the evaluation, only first evaluation set involve human judge as CMOS test, and it outperforms cosyvoice2 when the hard test, but not on the (ZH) easy test. There isn't subject evaluation on complex instruction-following or open instruction evaluation. As there are two reward model (Kimi-Audio-7B) and Emotion2Vec introduced in the GRPO, however, there are not subjective evaluation in the experiment which prove they are highly related to human perception.
>
> We appreciate the reviewer's comments on the evaluation methodology.
>
> **1. Subjective Evaluation on Complex Instructions**
>
> Conducting large-scale human evaluation for complex open-ended instructions is challenging due to the high variability in human interpretation. Therefore, we adopted the InstructTTSEval benchmark, where the authors have empirically demonstrated that **Gemini-based automated evaluation** exhibits a high correlation with human preference. Consequently, the results in Table 4, which rely on Gemini as the judge, serve as a reliable proxy for human perception in assessing complex instruction-following capabilities.
>
> **2. Validation of Reward Models**
> *   **Emotion2Vec**: This model is a state-of-the-art SOTA emotion recognition model pre-trained on large-scale datasets, widely recognized in the speech community for its alignment with human emotional perception.
> *   **Kimi-Audio-7B:** To validate Kimi-Audio as a reliable reward model, we evaluated its alignment with **Gemini-2.5-Pro** (the gold-standard judge) on InstructTTSEval. We measured the agreement (Macro-F1) of their judgments using the *Directly Judge* prompting strategy employed in our S3 training. As shown in the table below, Kimi-Audio achieves a solid agreement (Macro-F1 > 0.60) with Gemini. Given Gemini's proven correlation with humans, this result confirms Kimi-Audio’s validity as an effective training reward.
>
> | Metric                         | EN-APS | EN-DSD | EN-RP | ZH-APS | ZH-DSD | ZH-RP |
> | :---                           | :---:  | :---:  | :---: | :---:  | :---:  | :---: |
> | **Macro-F1 (Kimi vs. Gemini)** | 0.62   | 0.62   | 0.65  | 0.60   | 0.64   | 0.63  |

---

> ### Author Response · Authors · 2025-11-24
> **Response to reviewer Uu5e (Part 2/2)**
>
> > **[Question 1]** It seems the introduction of dataset for pretraining on FlexiVoice-Base. Paper mentions the FlexiVoice Instruct as 4316 hours which is used in pre-training stage. Does it mean it use the same corpus with the alignment stage?
>
> We would like to clarify the data usage across stages. The **FlexiVoice-Instruct** dataset (4,316 hours) is primarily used for pre-training. The subsequent alignment stages (post-training) utilize distinct data sources constructed for specific objectives, as detailed in Appendix A.4 and A.5:
>
> *   **S1 (Multi-modality DPO):** Uses paired preference data constructed from the ESD dataset to establish basic emotional alignment.
> *   **S2 (Decoupling GRPO):** Uses conflicting input scenarios constructed from NCSSD (text) and open-source emotion datasets (reference speech) to force disentanglement.
> *   **S3 (Instruction GRPO):** Uses a small subset of inputs from FlexiVoice-Instruct but heavily relies on 12,000 new instruction-text pairs generated by LLMs with varying difficulty levels to enhance complex instruction following.
>
> Thus, the alignment stages do not simply reuse the pre-training corpus but leverage targeted datasets to unlock specific capabilities.
>
> ---
>
> > **[Question 2]** Is there any quality comparison like between the FlexiVoice-Base and FlexiVoice? How many data pair used in the S1 DPO stage? And, if the pre-training stage add these instructions, how dose it compare with DPO alignment?
>
> **1. Quality Comparison (FlexiVoice-Base vs. FlexiVoice):**
>
> Yes, we provided comprehensive comparisons in Table 2 and Table 3 of the main paper.
> *   **Instruction Following:** FlexiVoice significantly outperforms the pre-trained model. For example, in Table 2 (EN-TO-Easy task), FlexiVoice achieves **97.4%** accuracy vs. **72.4%** for the Base model.
> *   **Audio Quality:** In Table 3, FlexiVoice shows comparable or slightly better subjective quality (Q-MOS: **4.08** vs. **3.72** in EN-TO-Easy), indicating that alignment improves controllability without degrading naturalness.
>
> **2. S1 DPO Data Size:**
>
> The S1 stage utilizes approximately 80,000 paired preference data samples constructed from the ESD dataset.
>
> **3. Pre-training vs. DPO Alignment:**
>
> Even if these instructions were added to pre-training (supervised fine-tuning), **DPO provides a distinct advantage**. Pre-training only maximizes the likelihood of the target speech given the instruction, whereas DPO explicitly optimizes the **preference margin** between matched (positive) and mismatched (negative) responses. This contrastive signal is crucial for forcing the model to strictly adhere to the instruction and ignore conflicting cues (e.g., emotional content in text), a capability that standard likelihood-based pre-training struggles to learn efficiently.

---

### Official Review · Reviewer_AyBD · 2025-11-01

**Soundness:** 3
**Presentation:** 2
**Contribution:** 2
**Rating:** 4
**Confidence:** 3

**Summary:**

The paper proposes a text-to-speech (TTS) system that can generate speech in any speaking style with any voice using natural language instructions for style control and reference speech for timbre control in a zero-shot manner. The model is built upon a  large language model (LLM) and trained using a novel Progressive Post-Training (PPT) scheme with three stages. The authors also introduce a new dataset, FlexiVoice-Instruct, annotated with natural language instructions via LLMs. Experiments on emotion datasets and the InstructTTSEval benchmark show FlexiVoice surpasses baselines (e.g., VoxInstruct, CosyVoice2) in style-timbre disentanglement, controllability, and naturalness.

**Strengths:**

1. This work bridges instruction-based and zero-shot TTS research, demonstrating how LLM-based reinforcement learning can generalize across diverse speaking scenarios. The progressive reinforcement learning curriculum (PPT) is novel in the context of TTS controllability. The unified design for style–timbre disentanglement with both natural language and reference inputs is a meaningful advance beyond previous instruction-based or zero-shot models.

2. This work addresses a critical and challenging problem in controllable TTS: achieving independent, flexible control over multiple attributes. The demonstrated ability to decouple style from timbre and content is a step forward.

3. The experimental demonstrates clear and substantial improvements over a wide array of strong baselines. The construction of the large-scale FlexiVoice-Instruct dataset is a major undertaking that adds considerable value to the community.

**Weaknesses:**

1. The decoupling evaluation is heavily focused on emotional control. While emotions are a key aspect of style, the paper would be strengthened by a more direct analysis of other stylistic aspects (e.g., speaking rate, pitch range, informal tone) to fully validate the "any style" claim. A qualitative analysis or case studies on non-emotional instructions would be valuable.

2. The progressive training scheme involving multiple stages of DPO and GRPO is computationally intensive. The paper does not report training time, GPU requirements, or model size, which are critical for replication.

3. Although Table 5 shows training-stage effectiveness, a finer-grained analysis of reward function contributions (DPO vs. GRPO weighting) would help interpret why specific stages contribute differently.

**Questions:**

Some issues and statements need to be clarified:

1. How does FlexiVoice perform on unseen speaking scenarios (e.g., narrative, dialogue) or in low-resource languages? How sensitive is the system to the balance between style fidelity and timbre preservation?

2. In Section 3.1, the authors mention using "Speak the following text" as a default instruction for data without explicit instructions. For the multi-modality tasks (TR) during inference, what is the precise input format when "only" a reference speech is provided without a style instruction? Does the model default to a neutral style, or does it attempt to infer a style from the reference?

3. In Section 3.2.3, you mention that for S3, you "discard references" to avoid conflict with open-ended instructions. However, in Appendix A.4, it is mentioned sampling "existing instruction-text inputs from the pre-training data." Please clarify if these pre-training data points included reference speech, and if so, how was this handled? Were the references simply omitted during S3 training?

4. The paper mentions that Kimi-Audio-7B was chosen over Gemini-2.5-Pro due to cost. Any validation is performed to ensure that the ranking or preference judgments from Kimi-Audio are sufficiently aligned with human preferences or with the more powerful Gemini model for your specific task?

---

> ### Author Response · Authors · 2025-11-24
> **Response to reviewer AyBD (Part 1/2)**
>
> Dear Reviewer AyBD,
>
> We sincerely thank the reviewer for the constructive feedback and have carefully addressed your concerns regarding the evaluation scope, reproducibility, and experimental analysis in our response below.
>
> > **[Weakness 1]** The decoupling evaluation is heavily focused on emotional control. While emotions are a key aspect of style, the paper would be strengthened by a more direct analysis of other stylistic aspects (e.g., speaking rate, pitch range, informal tone) to fully validate the "any style" claim. A qualitative analysis or case studies on non-emotional instructions would be valuable.
>
> We appreciate the suggestion to broaden the evaluation scope. We focused more on emotion for better understanding of the controllability since there are more resources on the evaluation of emotion. To further validate the "any style" claim beyond emotions, we conducted fine-grained control experiments specifically on **Speaking Speed** and **Pitch**. We do inference conditioned on instructions corresponding to three distinct levels (e.g., *Fast/Normal/Slow* for speed, *High/Normal/Low* for pitch) using instructions such as "Read this sentence quickly." We then calculated the Spearman Correlation Coefficient between the instruction levels and the extracted acoustic features (phoneme counts per second and median f0) to measure the model's instruction-following capability.
>
> As shown in the table below, FlexiVoice achieves the highest correlation scores, significantly surpassing baselines, demonstrating precise control over non-emotional prosody. Furthermore, we would highlight that the InstructTTSEval benchmark used in our main paper (Section 5.3) already explicitly evaluates a comprehensive set of stylistic attributes, including gender, pitch, clarity, fluency, speed, accent, age, etc. FlexiVoice’s superior performance on these complex tasks (Table 4) provides further evidence of its robust capability to handle diverse style instructions.
>
> | Model           | Speed (EN) | Speed (ZH) | Pitch (EN) | Pitch (ZH) |
> | :---            | :---:      | :---:      | :---:      | :---:      |
> | ParlerTTS       | 0.61       | 0.22       | 0.68       | 0.79       |
> | PromptTTS       | 0.24       | -          | 0.12       | -          |
> | PromptStyle     | 0.05       | -          | 0.24       | -          |
> | VoxInstruct     | 0.75       | 0.49       | 0.31       | 0.15       |
> | FlexiVoice-Base | 0.62       | 0.47       | 0.66       | 0.52       |
> | **FlexiVoice**  | **0.86**   | **0.78**   | **0.91**   | **0.90**   |
>
> ---
>
> > **[Weakness 2]** The progressive training scheme involving multiple stages of DPO and GRPO is computationally intensive. The paper does not report training time, GPU requirements, or model size, which are critical for replication.
>
> Thank you for highlighting this. FlexiVoice is based on Phi-3.5-mini-instruct (3.8b), and all training stages were conducted on 8 $\times$ NVIDIA A800 (80GB) GPUs.
>
> Regarding training costs, S1 is highly efficient, completing in approximately 2 hours (3 epochs, LR=1e-5, $\beta$=0.1). Two GRPO stages involve online sampling and run for 2 epochs each with LR=1e-5 and $\beta$=0.1. Specifically, S2 uses a group size of $G=8$ and takes ~36 hours, while S3 uses a group size of $G=6$ and takes ~42 hours. The total post-training pipeline requires approximately 3.5 days, which is manageable for a research-grade LLM-based TTS system.
>
> ---
>
> > **[Weakness 3]** Although Table 5 shows training-stage effectiveness, a finer-grained analysis of reward function contributions (DPO vs. GRPO weighting) would help interpret why specific stages contribute differently.
>
> We would like to take this opportunity to clarify the distinct roles of each stage.
>
> *   **S1 (Multi-modality DPO):** This stage serves as a **cold start** after pre-training. Relying on paired preference data, DPO provides strong supervision to activate the model's fundamental capability to process multi-modal inputs simultaneously. It establishes the initial alignment, ensuring optimization stability for subsequent stages.
> *   **S2 (Decoupling GRPO):** This is the key to **disentanglement**. By introducing conflicting data scenarios (e.g., happy instruction vs. sad reference) and optimizing explicit multi-objective rewards, we force the model to prioritize the instruction for style while maintaining speaker identity. This effectively suppresses style leakage from text content or reference speech, which DPO alone cannot fully address.
> *   **S3 (Instruction GRPO):** This stage addresses **complex generalization**. Since simple classifiers in S2 cannot evaluate open-ended instructions, we introduce a comprehensive ALM-based reward. This guides the model to handle diverse, real-world instructions.

---

> ### Author Response · Authors · 2025-11-24
> **Response to reviewer AyBD (Part 2/2)**
>
> > **[Question 1]** How does FlexiVoice perform on unseen speaking scenarios (e.g., narrative, dialogue) or in low-resource languages? How sensitive is the system to the balance between style fidelity and timbre preservation?
>
> **Generalization:** FlexiVoice demonstrates strong generalization to unseen scenarios. The InstructTTSEval benchmark (Table 4) explicitly evaluates tasks like "Role-Play (RP)" (dialogue) and "Descriptive-Style (DSD)" (narrative), where FlexiVoice achieves high performance, verifying its robustness in diverse contexts. Regarding low-resource languages, while our current experiments focus on English and Chinese, the underlying LLM (Phi-3.5) and pre-training corpus (Emilia) are multilingual, making the framework naturally extensible to other languages with appropriate data. We would like to investigate the generalization in low-resource languages in future work.
>
> **Sensitivity:** The system effectively balances style fidelity and timbre preservation without being overly sensitive to their trade-off. This robustness is a direct result of the **S2 (Decoupling GRPO)** stage, which optimizes both objectives simultaneously. Empirical evidence in Table 2 (TR-Hard task) shows that even in conflicting scenarios, FlexiVoice maintains high style accuracy while preserving high speaker verification scores, demonstrating that style expression does not come at the cost of timbre consistency.
>
> ---
>
> > **[Question 2]** In Section 3.1, the authors mention using "Speak the following text" as a default instruction for data without explicit instructions. For the multi-modality tasks (TR) during inference, what is the precise input format when "only" a reference speech is provided without a style instruction? Does the model default to a neutral style, or does it attempt to infer a style from the reference?
>
> In inference scenarios where only a reference speech is provided without a specific style instruction, FlexiVoice functions as a standard zero-shot TTS model. In this case, the model naturally clones both the timbre and the prosodic style (including emotion) embedded in the reference speech, rather than defaulting to a neutral style.
>
> ---
>
> > **[Question 3]** In Section 3.2.3, you mention that for S3, you "discard references" to avoid conflict with open-ended instructions. However, in Appendix A.4, it is mentioned sampling "existing instruction-text inputs from the pre-training data." Please clarify if these pre-training data points included reference speech, and if so, how was this handled? Were the references simply omitted during S3 training?
>
> Thank you for the opportunity to clarify. As stated in **Section 3.1** (Pre-training), the pre-training phase trains the LLM core "without incorporating the reference speech." Consequently, the "existing instruction-text inputs from pre-training data" sampled for S3 (as mentioned in Appendix A.4) inherently **do not contain reference speech**. Therefore, no reference speech was discarded or omitted during S3 training; it was simply not part of the input for these data points, consistent with the pre-training format. And they are only used for stable training process.
>
> ---
>
> > **[Question 4]** The paper mentions that Kimi-Audio-7B was chosen over Gemini-2.5-Pro due to cost. Any validation is performed to ensure that the ranking or preference judgments from Kimi-Audio are sufficiently aligned with human preferences or with the more powerful Gemini model for your specific task?
>
> To validate Kimi-Audio-7B as a reliable reward model, we evaluated its alignment with Gemini-2.5-Pro (the gold-standard judge in the InstructTTSEval benchmark) on the InstructTTSEval dataset. As shown in the table below, Kimi-Audio achieves a solid agreement (F1 > 0.60 across all tasks) with Gemini. This substantial alignment confirms that Kimi-Audio serves as a sufficiently accurate and cost-effective proxy in our instruction-speech consistency judgment task. Furthermore, the effectiveness of this reward model is empirically supported by Table 5 in our main paper, where the S3 stage (driven by Kimi-Audio reward) yields significant performance gains in open-ended instruction-following tasks, also reflecting the validity of the reward signal.
>
> | Metric                         | EN-APS | EN-DSD | EN-RP | ZH-APS | ZH-DSD | ZH-RP |
> | :---                           | :---:  | :---:  | :---: | :---:  | :---:  | :---: |
> | **Macro-F1 (Kimi vs. Gemini)** | 0.62   | 0.62   | 0.65  | 0.60   | 0.64   | 0.63  |

---

### Author Response · Authors · 2025-11-24
**General response to all reviewers**

Dear Reviewers,

We express our sincere gratitude to the reviewers for their thoughtful and constructive comments. Guided by your feedback, we have significantly revised the manuscript to clarify our contributions and strengthen our experimental validation. The updated sections are highlighted in **blue**. Below is a summary of the major improvements:

* **Title and Scope Refinement:** To address concerns regarding the "any style" claim being potentially overbroad, we have revised the title to "FlexiVoice: Enabling Flexible Style Control in Zero-Shot TTS with Natural Language Instructions". We have also adjusted related claims in the Abstract and Introduction to ensure the terminology reflects the system's flexible controllability more precisely.

* **Evaluation Beyond Emotion (Appendix A.8):** To validate the model's capability on non-emotional stylistic attributes, we conducted new fine-grained control experiments on Speaking Speed and Pitch. The results demonstrate that FlexiVoice effectively follows instructions for physical prosodic changes, significantly outperforming baselines.

* **Validation of Progressive Post-Training (Section 5.4 & Table 5):** To rigorously justify the necessity of our three-stage curriculum post-training, we added comprehensive ablation studies. We compared our approach against alternative training orders and joint training strategies. The results (Table 5) empirically confirm that the proposed progressive sequence is essential for balancing basic alignment (S1), disentanglement (S2), and complex generalization (S3).

* **Reproducibility and Reward Validation (Appendix A.7 & A.10):** We added a dedicated Reproducibility Details section specifying computational resources, training time, and hyperparameters. We also included an experiment validating our Kimi-Audio-based reward model against Gemini-2.5-Pro, showing strong alignment to ensure the reliability of our training signals.

Thank you again for your valuable feedback! We look forward to any further discussions and are happy to provide additional details if needed.

---

### Author Response · Authors · 2025-12-02
**Rebuttal Summary for Area Chair and Reviewers**

Dear Area Chair and Reviewers,

We thank the reviewers for their constructive feedback. As the rebuttal period concludes, we provide a summary of our core contributions and critical updates.

**I. Core Novelty and Strength**: Solving the "Style-Timbre-Content Conflict"

This paper addresses the most challenging problem in controllable speech synthesis: **achieving flexible style control via natural language while strictly preserving the timbre of a reference speaker**. Unlike previous works that struggle with modality entanglement (e.g., style leaking from text or reference speech), FlexiVoice introduces a novel Progressive Post-Training (PPT) framework.
- **Methodological Novelty**: We are the first to systematically adapt progressive Reinforcement Learning (DPO & Multi-objective GRPO) to enforce the disentanglement of style, timbre, and content.
- **Empirical Strength**: Our approach achieves state-of-the-art performance in instruction TTS tasks.
- **Community Contribution**: We will release FlexiVoice-Instruct, a large-scale (4k+ hours) dataset with diverse, LLM-annotated style instructions, filling a critical gap in the community. And also we will release FlexiVoice checkpoint and training code of our post-training strategy.

---

**II. Note on Reviewer Updates**

Due to the recent revert of system scores, the current ratings do not reflect the outcome of the rebuttal. We respectfully highlight that Reviewer Jm4v explicitly stated their intention to raise the score from 4 to 6 after reviewing our new ablation studies (see Official Comment by Reviewer Jm4v, 25 Nov 2025).

---

**III. Key Rebuttal Components**

Guided by the reviewers, we have strengthened the paper in three key dimensions:

1. Validated Methodological Necessity

> Concern: Reviewers (Jm4v, AyBD) asked for evidence justifying the necessity of the 3-stage progressive order compared to joint training or other sequences.

We conducted a comprehensive ablation study (added as Table 5 in the revision). The results empirically prove that: (1) Our PPT strategy significantly outperforms Joint Training (InstructTTSEval Avg 79.3 vs. 75.5). Joint training suffers from gradient conflicts between the strict constraints of S2 and S3. (2) Starting with complex task S3 first causes catastrophic failure (79.3 vs. 74.8), validating that the S1 "cold start" and S2 "disentanglement" are prerequisite foundations.

2. Refined Scope & Fine-Grained Evaluation

> Concern: Reviewers (AyBD, Uu5e) felt the original title ("Any Style") was overbroad and requested evaluations beyond emotional control.

We changed the title to "FlexiVoice: Enabling Flexible Style Control in Zero-Shot TTS with Natural Language Instructions" to precisely reflect our contribution. Additionally, we added experiments on Speaking Speed and Pitch control (see Appendix A.8 & Table 8). FlexiVoice achieved high Spearman Correlation Coefficients with the instruction (Speed: 0.86, Pitch: 0.91), significantly outperforming baselines like ParlerTTS and VoxInstruct. This proves the model generalizes to physical prosodic attributes beyond emotion.

3. Robustness & Reproducibility

> Concern: Reviewers requested more details on training costs and reward model validation.

We verified our Kimi-Audio-based reward model against the gold-standard Gemini-2.5-Pro, showing strong alignment (Macro-F1 > 0.60, Appendix A.7). We also added a detailed Reproducibility Details (Appendix A.8) with specific training costs (approx. 3.5 days on 8x A800) and hyperparameter details to facilitate future research.

---

We believe FlexiVoice represents a significant step forward in controllable, instructed zero-shot speech synthesis and hope the AC will consider these improvements in the final decision.

Best regards,

The Authors

---

### Meta-Review · Area_Chair_t3Xt · 2026-01-08

**Summary:**

The main concerns were whether the claimed “any style/any voice”is over broad since the training/evaluation was done on a small set of emotions, and whether the core novelty was truly the progressive three-stage post-training vs. simply using multiple objectives or known alignment methods. There are reproducibility concerns for such a computationally heavy pipeline, and questions about validating the disentangling of style–timbre–content.

**Reviewer Concerns:**

The rebuttal added an explicit ablation study showing that the progressive order and separation of stages are important, directly answering the “why progressive?” critique.  This led one reviewer raise their score to 6. The title/claims were reviewed and style evidence was broadened beyond emotion using speed/pitch experiments. Reproducibility was partially addressed by reporting more training details. Remaining concerns are mostly the strength-of-claim / depth-of-validation. Reviewers still do not see the progressive scheme as deeply insightful/novel, and the subjective evaluation is still largely proxy-based (Gemini judge) rather than new human studies.

**Reviewer Scores:**

Across the four reviewers, the scores seem to move from an average around 4.5 pre-rebuttal (three 4s and one 6) to an accept post-rebuttal. One reviewer explicitly shifts from 4 to 6 after the new progressive-vs-joint and order ablations. Others are likely to move up given the tightened claims and evaluation clarifications. Some issues remain even after fixes, as one reviewer frames the novelty as effective but not insightful, and the strongest validations for open-ended instruction-following still rely heavily on automated judging rather than expanded human studies.

---

### Decision · Program_Chairs · 2026-01-26

Accept (Poster)